# PAPER COPILOT: TRACKING THE EVOLUTION OF PEER REVIEW IN AI CONFERENCES

**Jing Yang** [*]
University of Southern California

**Qiyao Wei**
University of Cambridge

**Jiaxin Pei**
Stanford University

## ABSTRACT

The rapid growth of AI conferences is straining an already fragile peer-review system, leading to heavy reviewer workloads, expertise mismatches, inconsistent evaluation standards, superficial or templated reviews, and limited accountability under compressed timelines. In response, conference organizers have introduced new policies and interventions to preserve review standards. Yet these ad-hoc changes often create further concerns and confusion about the review process, leaving how papers are ultimately accepted—and how practices evolve across years—largely opaque. We present PAPER COPILOT, a system that creates durable digital archives of peer reviews across a wide range of computer-science venues, an open dataset that enables researchers to study peer review at scale, and a large-scale empirical analysis of ICLR reviews spanning multiple years. By releasing both the infrastructure and the dataset, PAPER COPILOT supports reproducible research on the evolution of peer review. We hope these resources help the community track changes, diagnose failure modes, and inform evidence-based improvements toward a more robust, transparent, and reliable peer-review system.

## 1 INTRODUCTION

The rapid growth of submissions to top-tier Artificial Intelligence (AI) and Machine Learning (ML) conferences—now exceeding 10,000 per venue annually (Yang, 2025)—has placed unprecedented pressure on peer review. To address issues of scale and transparency, many conferences have adopted open or semi-open platforms like OpenReview. Yet practices remain inconsistent: some venues publish reviews and scores, while others keep them private. This inconsistency has sparked ongoing debate around fairness, accountability, and the effectiveness of different review models (Cortes & Lawrence, 2021).

Meanwhile, the nature of review data itself has evolved. Whereas reviews were initially limited to a single numerical rating, nowadays reviews include multiple dimensions such as soundness, technical correctness, and presentation quality (Beygelzimer et al., 2021). Since the rise of OpenReview (Soergel et al., 2013) and especially following the widespread adoption of formal rebuttal phases, the need to analyze score dynamics over time has grown (Yang, 2025). Yet community discourse remains scattered—spread across Twitter, Reddit, Zhihu, Xiaohongshu, and other social platforms in fragmented, venue-specific threads. As submission volumes and review complexity continue to increase, the absence of structured tools that unify and visualize the review data has become a bottleneck for both transparency and timely author decision-making.

From the standpoint of conference organizers, internal dashboards provide visibility into score trends and acceptance distributions (Cortes & Lawrence, 2021). But these views are rarely made accessible to authors. During the critical post-review phase—often only one or two weeks long—authors are expected to analyze reviewer feedback, prepare rebuttals, and potentially decide whether to withdraw or revise their submissions. With scattered resources, authors must resort to scraping statistics (Sun, 2020) or aggregating posts from scattered social media. This inefficiency hinders timely decision-making and may even lead to missed opportunities for meaningful rebuttal or clarification. A system that actively collects review statistics, visualizes score dynamics, and contextualizes submissions relative to their peers would therefore offer immediate utility to thousands of authors.

---

[*]Corresponding Author, email `jingyang.carl.work@gmail.com`

In addition, while the OpenReview ecosystem has grown, many conferences still use closed-form review systems and opt not to release reviews or scores publicly. Motivations for withholding reviews may include protection of intellectual content, reviewer anonymity, or historical precedent. However, survey studies of peer review in ML have documented that such opacity exacerbates known problems: inconsistent reviewing standards, lack of calibration across reviewers, and limited accountability for low-quality or biased reviews (Shah, 2022). These shortcomings directly affect fairness and community trust, and they compound when authors have no visibility into aggregate statistics. A practical workaround is to allow voluntary, anonymized community submissions. If designed carefully, such a system can extract value from partially open data while respecting privacy and consent.

Beyond the review process itself, the AI / ML community lacks robust infrastructure for tracking who is shaping the field over time. Traditional platforms like Google Scholar and DBLP focus primarily on paper-level or author-level metadata—citations, references, and publication profiles (Scholar; DBLP). While Google Scholar does display author affiliations (and, by extension, country information), these systems were not designed to support temporal analysis or dynamic tracking across institutions or regions. Their scope remains limited, offering little visibility into broader affiliation-, institution-, or country-level dynamics. Tracking such information has historically been left to organizations due to the technical and logistical challenges involved, as seen in efforts like CSRankings (Berger) or university rankings.

As AI becomes increasingly high-profile and competitive, attention is increasingly concentrated around specific venue cycles. This shift highlights a growing need to understand who is actively driving progress over shorter timeframes: which institutions are rising, who remains active in the field, and what geographic regions are gaining influence. Yet no existing academic tool provides this kind of dynamic, multi-scale view. Ranking systems often update annually and draw from non-transparent data sources, making them ill-suited to capture the rapid and evolving dynamics of the field.

To address these needs and challenges, we introduce **Paper Copilot**, a system with open dataset for tracking peer review dynamics and talent trajectories across AI/ML. Our contributions include:

- **System for tracking AI/ML progress.** A venue-configurable pipeline (Fig. 1) that unifies multi-source inputs into standardized, versioned paperlists and powers interactive, multi-granularity analytics (venue/institution/country) for longitudinal progress tracking.
- **Scalable peer-review archive.** A unified archive built from open, semi-open, and opt-in community data with temporal snapshots, capturing multi-dimensional review metadata and daily dynamics (ICLR 2024/2025), to preserve and analyze cross-venue review evolution at scale.
- **Findings & ethical considerations.** Empirical results show a 2025 shift toward sharper, score-driven tiering under volume pressure and characteristic rebuttal-phase dynamics (score shifts, consensus evolution); we articulate safeguards on sourcing/consent, privacy, misuse, and bias.

## 2 RELATED WORKS

### 2.1 PEER REVIEW DATASETS AND ANALYSIS

There is a growing body of work on creating datasets to study academic peer review. PeerRead (Kang et al., 2018), collected 14.7K paper drafts with reviews and decisions from NLP / ML venues including ACL, NeurIPS and ICLR, enabling research on review–decision alignment and NLP applications like review score prediction. More recently, MOPRD (Lin et al., 2023) introduced a multidisciplinary open peer review dataset spanning several journals and computer science conferences, capturing the full reviewing process (including review comments, rebuttals, meta-reviews and final decisions) across domains. Other efforts have targeted specific aspects: for example, ORB (Szumega et al., 2023) curated a list of more than 36,000 scientific papers with their more than 89,000 reviews and final decisions in high energy physics.

Bharti et al. (2022) leveraged an NLP approach to estimate reviewer confidence from review text, illustrating analysis that can be done when reviews are available. Peer Review Analysis (Ghosal et al., 2022) provided 1,199 review reports labeled at the sentence level for aspects like critique, suggestion,

and sentiment, serving as a benchmark for automated review assessment. Ribeiro et al. (2021) used a private dataset of two non-disclosed conferences to perform RSP, PDP, and sentiment analysis to extract review polarities. In general, these datasets have been invaluable in enabling studies on review quality, bias, and consistency. However, each is often limited to a particular venue or a snapshot in time. We advance beyond prior datasets by unifying multiple venues over a long temporal window, thereby supporting cross-conference and longitudinal analyses (e.g., how review score distributions shift over years, or how an author's successive submissions fare over time). It also introduces the notion of tracking review process "dynamics," since our data include timestamps and sequences (allowing one to reconstruct the timeline of reviews and discussions for open-review conferences). This complements earlier one-off studies that examined review consistency and randomness in single years (Beygelzimer et al., 2021; Cortes & Lawrence, 2021), by providing a broader data foundation to study such effects in the aggregate.

## 2.2 Peer Review Tracking Systems

Our work is related to platforms that manage or expose the peer review process. OpenReview.net is the most prominent system enabling open peer review for many ML conferences (ICLR, NeurIPS, UAI, etc.), providing public APIs to fetch submissions, reviews, and comments (OpenReview, 2024). We extensively leverages OpenReview as a data source, but whereas OpenReview is focused on facilitating the review process for active conferences, our system is designed to track and analyze review outcomes across conferences and years. In some sense, Paper Copilot serves as a meta-layer on top of systems like OpenReview and other open-access portals and proceedings (Foundation, 2024; Anthology), aggregating their outputs for comparative analysis. A few individual conferences have released summary statistics (e.g., acceptance rates, score distributions) in blog posts or slides, but these are ad-hoc and not standardized (Beygelzimer et al., 2021). To our knowledge, there was no public web resource before Paper Copilot that allowed users to conveniently browse and compare peer review results (scores, decisions, etc.) for a wide range of AI / ML venues in one place. Our system fills this gap by acting as a centralized peer review statistics dashboard for the community. It is somewhat analogous to "CS Conference Stats" websites that track acceptance rates (Berger), but we go further by including detailed review metrics and providing the underlying dataset.

## 2.3 Author Profiling and Metascience

Studying author career trajectories and publication patterns has been a long-standing interest in bibliometrics (Fan et al., 2024; De Frutos Belizon et al., 2025). Large-scale academic databases and knowledge graphs – such as Google Scholar, Microsoft Academic Graph, Semantic Scholar's Open Research Corpus, and AMiner – have enabled analyses of collaboration networks, citation trajectories, and researcher mobility (Scholar). For example, AMiner (Tang et al., 2008), an academic social network mining system that compiled tens of thousands of author profiles and their affiliations over time, allowing for queries on career moves and productivity. A recent dataset from AMiner contains over 57,000 scholars' career paths (with 42,230 affiliations) for studying academic job transitions. These tools, however, focus on publication and citation data; they generally do not incorporate the peer review stage. We brings a new dimension by linking authors to the review process outcomes of their submissions. This creates opportunities for "talent trajectory" studies in AI / ML – for instance, one could track how an early-career researcher improves their paper quality (as reflected in review scores) over successive submissions, or examine whether consistent high reviewers' ratings correlate with future citation impact (Vinkenburg et al., 2020). Our dataset, combined with external bibliometric data, could also help identify patterns such as mentorship lineages (e.g., how students of certain groups perform in reviews) or geographic trends in research feedback (Guevara et al., 2016). In the broader context of metascience, our work contributes an open resource to empirically study questions about peer review and researcher development. This aligns with recent calls for transparency and data-driven policy in peer review (Bianchi & Squazzoni, 2022). By making our data easily accessible, we hope to facilitate further meta-research on how review processes influence and reflect the growth of the AI / ML research community.

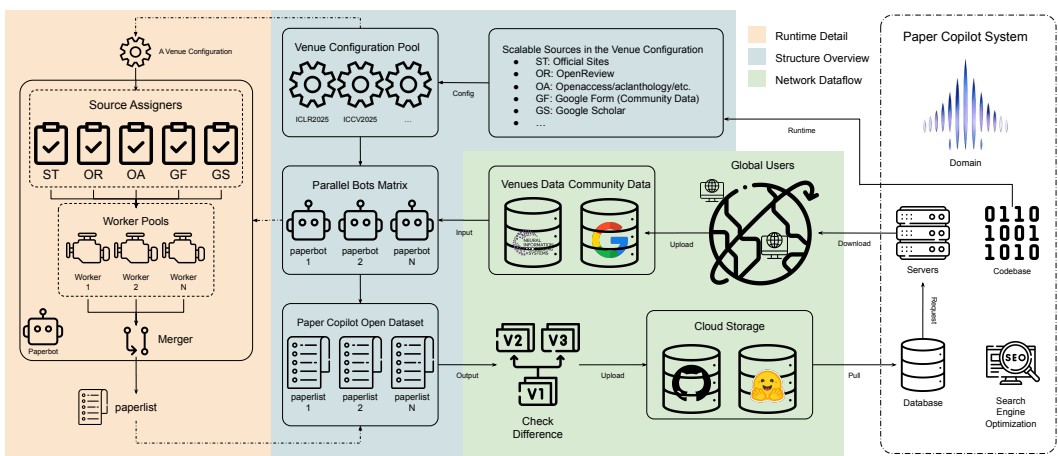

Figure 1: Paper Copilot System Architecture

## 3 SYSTEM

Paper Copilot is a modular system for large-scale collection and presentation of peer review dynamics and talent trajectories. Figure 1 outlines its architecture: venue configurations define scalable sources, processed through assigners, worker pools, and parallel bots to generate standardized paperlists. Versioned datasets are validated, uploaded to cloud storage, and served globally through optimized backend, database, and frontend components. The following sections detail data collection, system backend/frontend, and maintenance.

**Data Collection**  We use a hybrid data ingestion pipeline to accommodate diverse conference policies, including: 1) *OpenReview API*: For open-review venues (e.g., ICLR, NeurIPS), scheduled scripts pull submission metadata, review scores, confidences, and discussion threads. We store timestamped snapshots to track score changes (e.g., pre-/post-rebuttal). 2) *Website Scraping*: For venues without APIs (e.g., CVPR, AAAI), lightweight scrapers extract accepted papers, authors, and other publicly available metadata from official websites or proceedings. 3) *Community Contributions*: For closed-review venues, authors voluntarily submit reviews via a structured form, yielding 6,584 valid reviews per submission to date.

All data flows through a cleaning and normalization module that standardizes fields, deduplicates entries, and prepares them for integration. Adding new venues requires minimal configuration.

**Backend and Frontend**  The backend is built on a LAMP stack (Linux, Apache, MySQL, PHP) and deployed via a cPanel-managed hosting environment, which simplifies cron scheduling, backups, and database management. Conference data—submissions, reviews, authorship—is stored in a MySQL relational database with a normalized schema linking papers, reviews, and authors. Pre-computed aggregates (e.g., average score, acceptance rate per venue/year) are cached to support fast retrieval for analytics and display.

The user-facing website is built with WordPress, extended by custom PHP and JavaScript for dynamic rendering. All pages support filtering by score range, decision status, or specific metadata fields, and provide links to external resources such as OpenReview, official proceedings, or ArXiv when available. Each conference currently has two primary views: 1) *Statistics*: Interactive charts display trends in paper counts, acceptance rates, score distributions, and reviewer score rankings. 2) *Paper list*: Tabulated entries show paper titles, primary areas, authors, affiliations, departments, countries, citations, and acceptance types (e.g., oral/poster).

Beyond individual venues, the system supports cross-venue, longitudinal analysis at the institutional level. Users can explore hierarchical structures—tracking contributions over time from institutions to departments, laboratories, and regions. This enables global and fine-grained insights into talent flows, geographic shifts, and institutional participation across the field. For more advanced exploration, we provide a companion search tool that can run locally. Users can query by score thresholds, keywords, or metadata fields, operating on either the live database or JSON exports.

**Maintenance and Updates** The entire Paper Copilot platform—from backend to frontend—is fully automated and designed for scalability, reliability, and long-term meta-research with minimal manual intervention. Cron jobs manage regular updates to review dynamics; scraper failures trigger alerts; and the database is routinely backed up to ensure data integrity. Static pages are cached to reduce server load, and the modular pipeline allows easy adaptation as conference policies evolve. Developed and maintained at low cost, the system's modular architecture enables rapid scaling to new venues with minimal overhead. All processed datasets are publicly available, as detailed in Section 4.2, supporting transparency, reproducibility, and community-driven enhancements to features such as search and visualization.

## 4 DATASET

The Paper Copilot open dataset spans decades of peer review data and accepted paper metadata across dozens of AI / ML conferences and continues to grow. In this section, we detail the dataset's contents, structure, and coverage. We describe how temporal review profiles are represented, the coverage of authors and venues, and provide examples of insights one can derive. We also explain the format in which we release the data and the visualizations included.

### 4.1 PEER REVIEW DYNAMICS

**Review Dimensions** As AI / ML research expands, review criteria have grown in both number and complexity, varying across venues and years. In addition to core metrics like rating and confidence, many conferences now assess dimensions such as soundness, correctness, novelty, contribution, presentation, etc. These evolving schemas reflect a shift toward more granular peer evaluation.

**Reviewer-Author Discussion Score Dynamics** For venues with fully open review processes (e.g., ICLR), we leverage the public API to track the temporal evolution of review scores, confidence, and comments throughout the discussion and rebuttal phases. This allows us to construct reviewer profiles over time and analyze how feedback shifts during the review cycle. By timestamping each review snapshot, our system captures pre- and post-rebuttal score changes at the individual reviewer level, enabling fine-grained analysis of reviewer behavior and consistency. Figure 3 illustrates these dynamics using ICLR 2025 public data. Although ICLR reviews are public, historical snapshots are not preserved on the official platform—older versions are overwritten during the discussion phase. Paper Copilot continuously archives these updates in real time, providing what is, to the best of our knowledge, the **only publicly accessible archive of review score dynamics available anywhere on the internet**.

For closed-review venues that do not disclose review revisions (e.g., CVPR, ICCV, ICML), we invite authors to voluntarily submit their initial and final review scores. This opt-in process enables partial reconstruction of rebuttal impact, even when the official platform lacks transparency. By combining automated tracking and community-contributed data, our system enables venue-level comparisons of rebuttal effectiveness and decision volatility across time.

**Closed Venue Review Disclosure** Due to the closed nature of most conference reviews, public access to detailed review data remains limited. As part of our community engagement, we launched an opt-in questionnaire alongside our score collection process to assess author willingness to anonymously share their reviews. This study was conducted across major venues, including CVPR, ICML, ICCV, and ACL, and will continue with NeurIPS 2025.

To characterize community attitudes toward transparency, we conducted a user study across four major conferences in 2025. In total, we collected 1,860 responses, of which 1,115 authors (59.9%) consented to publicly release their anonymized review scores. Consent rates were broadly consistent across venues: 53.5% at CVPR (191 of 357), 60.7% at ICML (628 of 1,034), 59.4% at ICCV (151 of 254), and 67.4% at ACL (145 of 215). These results indicate strong support—roughly 60% overall—for releasing anonymized review data. We plan to continue this initiative across future years and venues to build a longitudinal dataset that can inform empirical studies of peer review models.

## 4.2 DATASET FORMAT AND RELEASE

To ensure usability and reproducibility, we release the dataset in structured JSON format, with one record per paper containing its metadata, review scores, timestamps, and author/affiliation information. Each paper record includes over 30 fields covering metadata, authorship, review scores, rebuttal dynamics, and final decisions. More details of data acquisition can be found in the supplementary material. The main dataset is available on GitHub A separate repository, GitHub, hosts the temporal data with parsing code for tracking review dynamics over time, due to size constraints. We are also collecting ICLR 2026 review data, which will be integrated into the archive in future updates if the conference continues to adopt a fully open review model.

## 5 THE EVOLUTION OF PEER REVIEW IN AI CONFERENCES.

In this section, we analyze the evolution of peer review in ICLR, a frontier AI conference. Our analysis aims at two key questions: (1) how has the conference decision-making mechanism evolved across years and (2) what are the discussion dynamics of ICLR during the rebuttal period?

### 5.1 THE EVOLUTION OF CONFERENCE DECISION-MAKING MECHANISMS

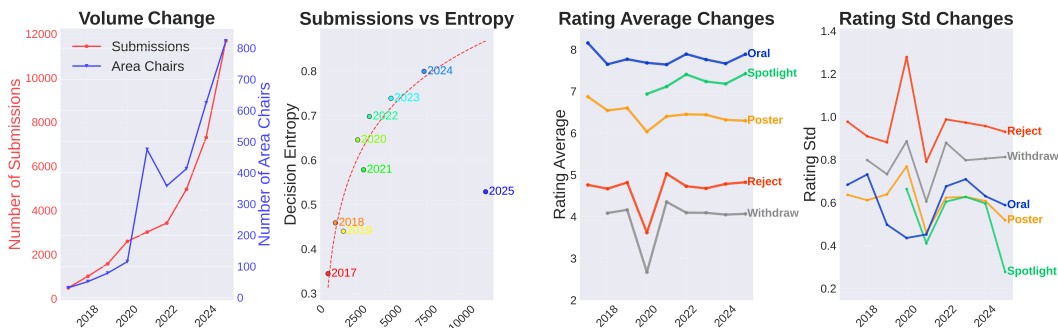

Figure 2: ICLR review dynamics from 2017 to 2025. Despite stable score distributions, decision entropy rises with scale—until 2025, where it drops sharply. This deviation suggests Area Chairs play a more decisive role, increasingly relying on mean scores under high submission pressure.

Over the past decade, ICLR has experienced explosive growth—from just 490 submissions in 2017 to 11,672 in 2025 (Figure 2, leftmost panel). To manage this scale, the number of Area Chairs (ACs) expanded proportionally, rising from 31 to 823 over the same period. This rapid expansion underscores the systemic pressure imposed by volume growth.

When analyzing the **final decisions** made by ACs, we quantify uncertainty using **decision entropy**, which measures unpredictability in tier assignment given a paper's mean score. Formally, for year $t$ and score bin $b$:

$$H_{t,b} = - \sum_{s \in \{\text{Reject, Poster, Spotlight, Oral}\}} p_{t,b,s} \log p_{t,b,s}, \qquad \bar{H}_t = \sum_b w_{t,b} H_{t,b},$$

where $p_{t,b,s}$ is the empirical probability of status $s$ for papers with binned mean score, and $w_{t,b}$ are bin-level weights. Higher $\bar{H}_t$ indicates greater uncertainty; lower values indicate sharper, score-driven decisions.

As shown in Figure 2 (center-left), $\bar{H}_t$ typically increases with submission volume $X_t$, and is well approximated by a logarithmic scaling:

$$\bar{H}_t \approx a \log X_t + b.$$

This trend aligns with an ordered-logit model for score-to-status mapping:

$$P(\text{status} = s \mid x) = \sigma(\tau_s - \kappa_t x) - \sigma(\tau_{s-1} - \kappa_t x),$$

where $\sigma$ is the logistic CDF and $\kappa_t$ reflects decision sensitivity to mean score in year $t$. As $X_t$ grows, $\kappa_t$ typically softens, yielding higher entropy.

Yet **ICLR 2025 is a clear outlier**. Despite the largest submission volume ($X_{2025} = 11,672$), $\bar{H}_{2025}$ fell below the fitted log trend. This implies a stronger $\kappa_{2025}$, i.e., the venue relied more deterministically on average scores, reducing uncertainty rather than increasing it. The residual,

$$\text{resid}_{2025} = \bar{H}_{2025} - (a \log X_{2025} + b),$$

is strongly negative, showing that scale alone cannot explain the drop.

Turning to score allocation (Figure 2, center-right and rightmost panels), **average ratings** have grown more tier-separated, with Spotlights converging toward Orals. Meanwhile, **rating variance** rises for rejected/withdrawn papers but drops for accepted ones, especially Spotlights. This shows ACs increasingly favor high and stable scores, sharpening thresholds and limiting borderline flexibility.

Together, these results indicate a structural change in 2025: ACs assumed a more decisive role, enforcing sharper score-dependent rules. The system shifted from probabilistic tiering to near-deterministic mappings, improving efficiency but potentially narrowing the role of reviewer justifications, rebuttals, or confidence.

## 5.2 Discussion Dynamics

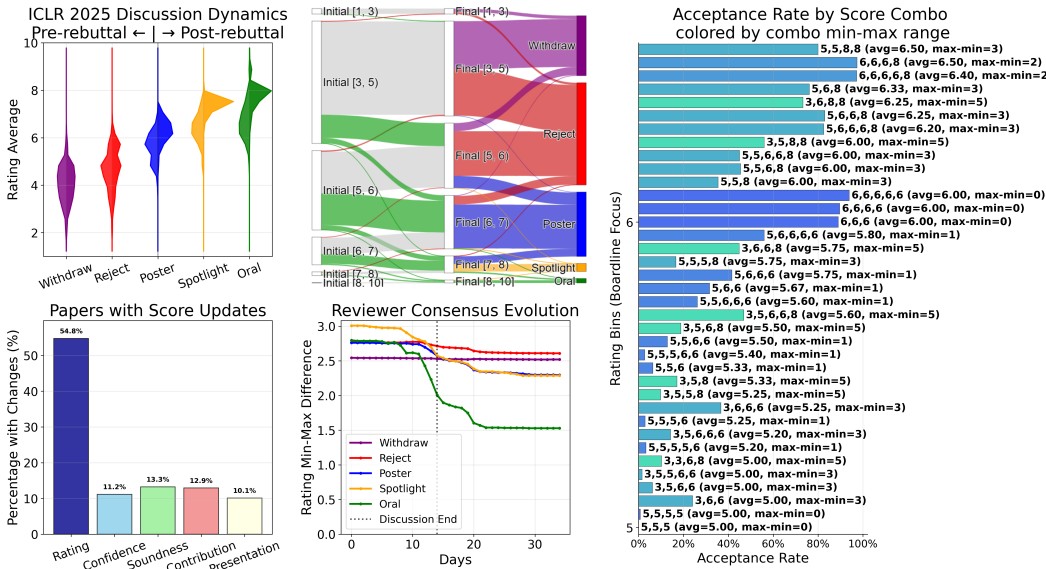

Figure 3: **Rebuttal dynamics in ICLR 2025**. (top-left) Ridge plots of pre- vs. post-rebuttal ratings across final decision tiers. (top-mid) Sankey plot tracking score trajectories from initial rating bins to final bins and final statuses. (lower-left) Distribution of score updates across review dimensions. (lower-mid) Reviewer consensus evolution measured by min–max rating difference. (right) Acceptance rates conditioned on reviewer score combinations with frequency cutoff=30.

To complement the volume and entropy analysis, we examine how the rebuttal phase shaped reviewer behavior and final decisions in ICLR 2025 (Figure 3).

**Score shift before and after rebuttal** The ridge plot (top-left) shows that score shifts after rebuttal were most pronounced in higher tiers. Spotlight and Oral papers experienced clear upward movement, whereas Withdrawn and Rejected papers showed little change. The Sankey diagram (top-right) further traces the evolution of scores from initial assignments to post-rebuttal updates and final status. Here, green flows denote upward shifts in mean score and red flows denote downward shifts. The dominant pattern is that many borderline and mid-tier papers received upward adjustments, while downward movements remain less common but non-negligible.

**Reviewer score change patterns and consensus** At a finer granularity, score updates were not uniform across review dimensions (bottom-left). Over half of papers (54.8%) saw changes in the overall rating, whereas dimensions such as confidence, soundness, contribution, and presentation

shifted in only ∼10–13% of cases. This suggests that rebuttals primarily influenced holistic ratings rather than specific dimension scores.

Reviewer consensus dynamics (bottom-center) show that rating min–max differences consistently narrowed as discussions unfolded. Notably, consensus levels dropped sharply at the onset of the discussion phase (marked by the vertical dashed line), reflecting the divergence that arises when reviewers actively debate rebuttal content. Toward the end of the process, consensus recovered, with Oral papers converging most strongly by the decision deadline, while Rejected papers retained higher levels of disagreement. This indicates that high-stakes papers underwent deeper discussion and alignment, while weaker papers received less effort in consensus-building.

**Acceptance rate and score patterns**  Finally, we inspect acceptance rates at the borderline using reviewer score combinations (bottom-right). After filtering for frequency (cutoff = 30), we find a notable asymmetry: when the average score is slightly above the borderline, lower min–max ranges correlate with higher acceptance rates; conversely, when the average is slightly below, higher min–max ranges correlate with acceptance. A plausible explanation is that below the borderline, a single strongly positive reviewer can sway the outcome, while above the borderline, persistent disagreement often signals unresolved concerns leading to rejection. While compelling, this empirical pattern warrants further validation in future analyses.

Taken together, these findings highlight the dual role of rebuttals: amplifying score changes for borderline cases and driving consensus formation for stronger papers. The corresponding plots for ICLR 2024 are provided in the appendix, enabling year-over-year comparison.

## 6  ETHICAL CONSIDERATIONS

The development of a dataset tracking peer review and talent trajectories necessitates a careful and transparent approach to ethical considerations. We are committed to responsible data stewardship and have outlined our mitigation strategies for key ethical challenges below.

### 6.1  DATA SOURCING, LICENSING, AND CONSENT

One potential concern revolves around the methods of data collection and adherence to licensing and Terms of Service (ToS). Our data is aggregated from three distinct sources, each with a different consent and licensing model. 1) **Public APIs and Data Sources:** A significant portion of our data is sourced from platforms with explicit data-sharing policies, such as OpenReview, which provides a public API for accessing review data. Data from such sources is collected in accordance with their established terms. 2) **Web Scraping:** Automated scraping can violate the ToS of some platforms, such as Google Scholar. To ensure full compliance, we have audited our data collection pipelines and ceased all automated data collection that conflicts with site 'robots.txt' files or explicit ToS. 3) **Community-Contributed Data:** For closed-review venues, we rely on voluntary, opt-in submissions from authors. Each author who contributes data does so via a form that includes explicit consent questions. Authors can choose whether their submission contributes only to anonymized, aggregate statistics or if their anonymized scores can be displayed in detailed tables showing individualized statistics.

### 6.2  PRIVACY, ANONYMITY, AND RE-IDENTIFICATION RISK

Another concern, potentially related to data licensing, is about protecting the identities of both authors and reviewers.

We do not collect, store, or attempt to de-anonymize reviewer identities. The vast majority of review data comes from platforms like OpenReview, where reviewer identities are already anonymized by default. There are potentially concerns that author trajectories, especially when combined with public timelines or multiple affiliations, could create re-identification risks. To mitigate this, our primary analysis focuses on **large-scale, aggregated trends at the institutional and geographic levels rather than individual-level data**. In addition, we are committed to implementing privacy-preserving measures, such as applying differential privacy to aggregate statistics.

### 6.3 POTENTIAL FOR MISUSE AND DUAL-USE RISKS

The dataset's potential for misuse, particularly in hiring and evaluation, is a significant "dual-use" risk. A worst-case scenario involves the data being used to unfairly judge job candidates, especially early-career researchers. Our stated goal is to increase transparency in the peer review process and empower researchers with a broader view of the academic landscape, not to create a performance ranking tool. To guard against misuse, we will take the following steps: 1) *Explicit Use Guidelines:* The Paper Copilot website will feature prominent guidelines strongly discouraging the use of its data for hiring, promotion, or other high-stakes evaluations of individuals. 2) *Focus on Aggregates:* The platform will prioritize the visualization of aggregated, large-scale trends over individual-level data. 3) *Community Dialogue:* We are committed to engaging in an ongoing dialogue with the AI/ML community to monitor how the data is used and to develop stronger safeguards as needed.

### 6.4 DATA INTEGRITY, INACCURACY, AND BIAS

The dataset is subject to several sources of potential inaccuracy and bias.

**Sampling and Representation Bias.** For closed-review venues, the reliance on community submissions introduces a self-selection bias. While this is currently the only feasible method for tracking review dynamics at scale without official data access, we recognize its limitations. To rigorously quantify this sampling bias, we plan to conduct a comparative analysis between our community-submitted data and the official ground-truth data from NeurIPS 2025 once it is released.

**Data Inaccuracy.** Inaccurate data, whether from parsing errors or malicious submissions, could harm researchers' reputations. The LLM-based affiliation extraction has a non-zero error rate. We will mitigate this by: 1) Clearly labeling the source of all data (e.g., "Official API", "Community-Submitted"). 2) Providing transparent metrics on the known error rates of our extraction models. 3) Implementing validation checks for community submissions to flag anomalous entries.

**Demographic and Geographic Bias.** The dataset could contain demographic and geographic biases, posing a special risk to researchers from certain groups (e.g., by geography, gender, or race). This bias can manifest in several ways. For closed-review conferences, the dataset's reliance on voluntary community submissions may skew it toward certain regions or institutions that are more culturally inclined to share data. Furthermore, some research groups have internal policies that discourage sharing review data even when it is possible to opt-out, which could lead to the systematic underrepresentation of those groups in our dataset.

Such imbalances could lead to skewed analyses, where trends observed in overrepresented groups are incorrectly generalized to the entire research community, potentially disadvantaging those in underrepresented regions. To actively address this, we are committed to the following actions:

- **Transparency via Datasheets:** We will publish and maintain a datasheet that transparently documents the known geographic and institutional distributions in our dataset, highlighting potential skews so that researchers can interpret the data with proper context.
- **Regular Bias Audits:** We will perform regular audits to quantify these biases. Where possible, we will compare the distribution in our data against available ground-truth statistics from conference organizers to measure the extent of the disparity.
- **Platform-Level Context:** On our platform, any analyses derived from potentially biased data sources, such as community submissions, will be clearly marked with disclaimers that explain the data's limitations and advise against overgeneralization.

## 7 CONCLUSION

Paper Copilot provides a unified, large-scale platform for tracking peer review dynamics and talent trajectories across major AI / ML conferences. It enables comparative analysis of score trends, rebuttal effectiveness, and institutional influence, with datasets already adopted by both academia and industry. Key limitations include partial reliance on voluntary data contributions for closed-review venues and the need to further improve the accuracy of talent trajectory reconstruction. Future work

will focus on expanding venue coverage and enhancing metadata quality for more comprehensive and precise analysis.

**Scope and Clarifications.** The core contribution of this work lies in the combination of a standardized multi-source infrastructure for peer-review data collection and normalization, the ICLR Daily Snapshot dataset that preserves temporal review states otherwise overwritten by existing platforms, and the empirical analysis enabled by this record. Although the system is publicly deployed through a web interface, the scientific contribution is not the interface itself, but the underlying archive, data pipeline, and metascientific analysis. All main quantitative results in this paper are derived exclusively from official ICLR data; community-contributed data remains explicitly separated, clearly labeled, and is not used to support the paper's scientific claims. We further emphasize that statistics such as entropy are used here as descriptive measurements of review dynamics rather than as definitive causal evidence, and multiple interpretations remain possible, including both review-process pressure and improved consistency in evaluation. The system was also designed to extend naturally to ICLR 2026, and we continue to archive its score evolution; however, because the ICLR 2026 review cycle encountered an OpenReview reviewer-information leakage issue, we treat the 2026 record in this work only as a reference signal rather than as part of the main analysis. Finally, broader capabilities of the system, such as tracking talent trajectories across institutions and years, are part of the extensible scope of the platform but are not presented here as current empirical claims.

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
