# Supplementary Material

## A Statement

### A.1 The Use of Large Language Models (LLMs)

Per the conference author guidelines, we disclose our use of large language models (LLMs). We used LLMs as general-purpose assistants to improve readability and perform light proofreading (grammar, phrasing, consistency), and to support data analysis in limited ways (e.g., drafting exploratory scripts/snippets, suggesting plotting templates, flagging potential edge cases, and summarizing intermediate results). The paper's structure, scope, and content were devised by the authors after discussions with coauthors. All analyses, statistical choices, model specifications, and interpretations were decided and verified by the authors; LLMs did not autonomously design experiments, select results, or generate data/tables/figures. The authors take full responsibility for all content. LLMs are not eligible for authorship, and their usage here does not constitute contribution beyond editorial and tooling assistance. This disclosure follows the venue's policy requiring a dedicated section when LLMs are used; nondisclosure of significant use may result in desk rejection.

### A.2 Reproducibility Statement

Paper Copilot is a live, continuously updated system; exact replication of the website state at any given time is therefore not possible. To ensure reproducibility of the results reported here, we will release (i) a versioned snapshot of the datasets used in all analyses, and (ii) companion code that deterministically regenerates every figure and table from this snapshot.

Our data acquisition pipeline includes community-contributed records. For privacy, consent, and licensing reasons, we do not plan to fully disclose raw records or the end-to-end processing pipeline. Instead, we provide an anonymized and aggregated dataset sufficient to reproduce all reported statistics, which can be cross-verified against publicly accessible online sources at the snapshot time (e.g., official conference exports and archival pages). In parallel, we are developing bias-correction procedures; to enable ongoing audit and extension, we plan to open-source a repository that implements these modules and documents their assumptions.

## B More Visuals and Analysis

### B.1 ICLR Dynamics

Figures 1 and 2 summarize cross-year patterns in confidence and rating. Rating distributions increasingly separate by final tier: Oral/Spotlight shift upward while Reject/Poster stabilize lower, and volume accumulates in mid–high rating bins (5.6–7.0) as scale grows. Confidence, in contrast, remains comparatively stable across years and tiers, with medians clustered around 3.2–4.0 and only modest widening for high tiers. Status–year grids (pies) show stronger stratification by rating than by confidence, while volume circles highlight that growth first concentrates near the decision boundary.

**Findings.** (1) *Ratings dominate outcomes*: above $\sim 7.4$, status pies become increasingly pure Oral/Spotlight; below $\sim 6.0$, Reject dominates across years. (2) *Borderline band*: the largest submission growth lies in 5.6–6.6, where acceptance has trended upward, consistent with sharper AC thresholds under load. (3) *Confidence is secondary*: at fixed confidence, status mixtures change little; high confidence does not imply acceptance without high rating. (4) *Scale effects*: densification in mid–high rating bins suggests competition near the boundary, reinforcing score-driven, more deterministic decisions. Overall, confidence modulates certainty, whereas rating governs tier assignment.

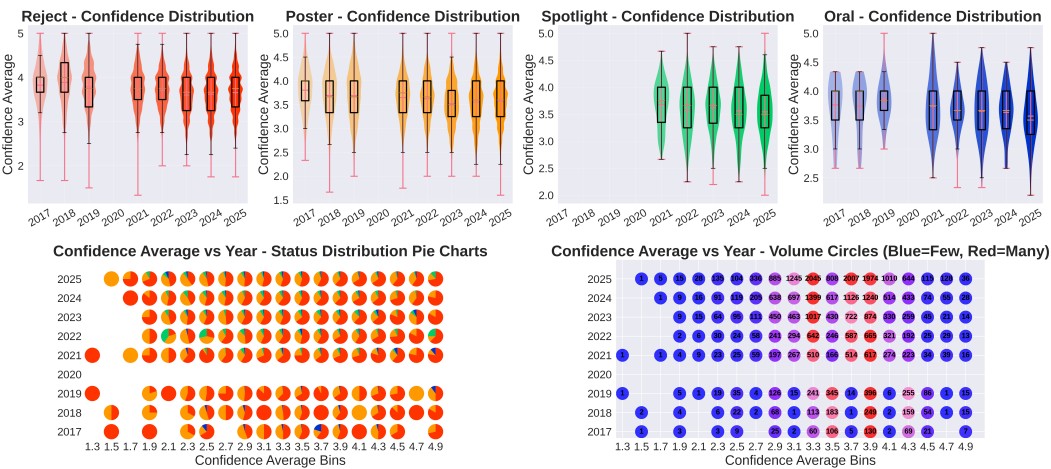

Figure 1: Confidence-focused view: per-tier confidence distributions (top) and confidence–year grids showing status mix (pies) and volume (circles) by confidence bins (bottom).

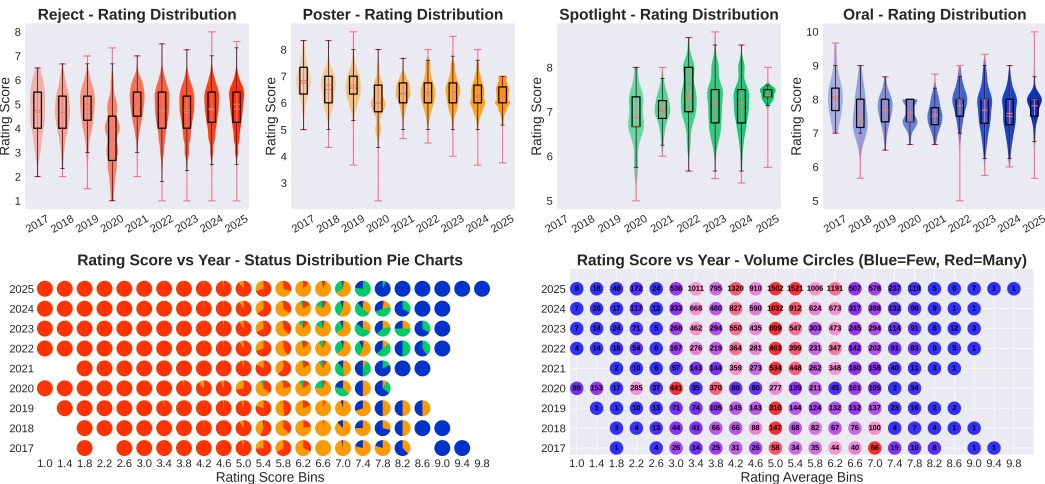

Figure 2: Rating-focused view: per-tier rating distributions (top) and rating–year grids showing status mix (pies) and volume (circles) by rating bins (bottom).

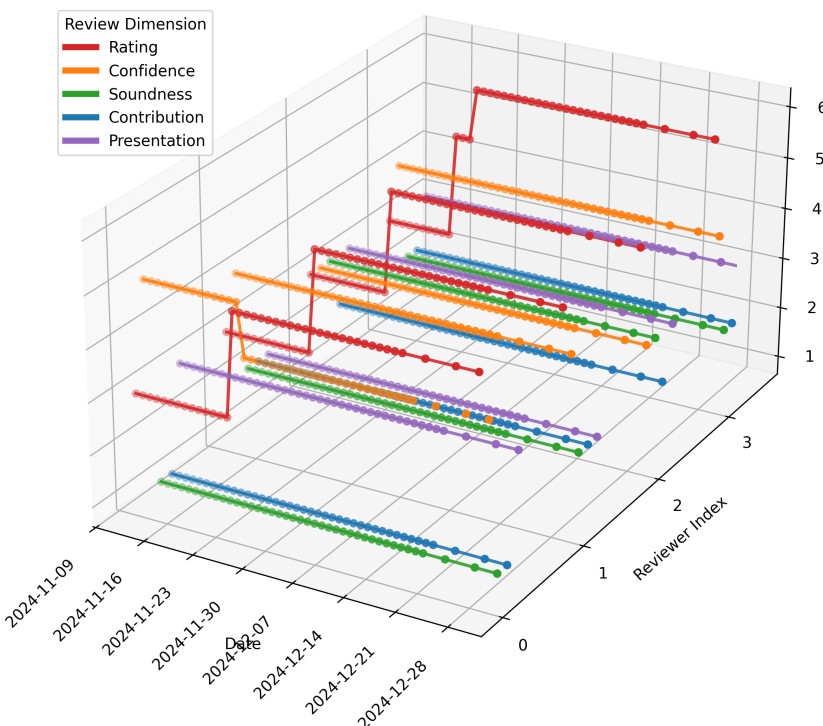

Figure 3: Example score footprint showing multi-dimensional score updates across time and reviewers.

## B.2 Score Footprints

We represent a sample paper's review trajectory as a **score footprint** (Figure 3), capturing how multiple dimensions (rating, confidence, soundness, contribution, presentation) evolve across time and reviewers. This view reveals temporal updates (e.g., post-rebuttal shifts), asynchronous changes across dimensions, and heterogeneous reviewer behaviors. The footprint provides a structured interface for analyzing dynamics of consensus and decision-making.

## B.3 ICLR 2024 vs ICLR 2025

Figure 4 and Figure 5 contrast discussion dynamics across two years. Several differences emerge:

**Similarities**. In both years, score updates concentrate heavily on the overall rating dimension, and consensus generally improves as the discussion progresses. Moreover, acceptance rates remain strongly tied to mean score, with low-scoring papers rarely advancing regardless of variance.

**Differences**. In 2025, the fraction of papers with score updates increased substantially (from below half to over 50%), and revisions extended across more dimensions (confidence, soundness, presentation). Reviewer consensus also converged more sharply, particularly for high-tier outcomes, reflecting stronger AC intervention. Finally, borderline score bins that were only moderately accepted in 2024 saw noticeably higher acceptance in 2025, indicating a shift toward sharper thresholds.

Overall, while both years show the same structural dynamics, ICLR 2025 displays a stronger reliance on score-driven and consensus-enforced decision rules, contrasting with the more gradual and distributed adjustments of 2024.

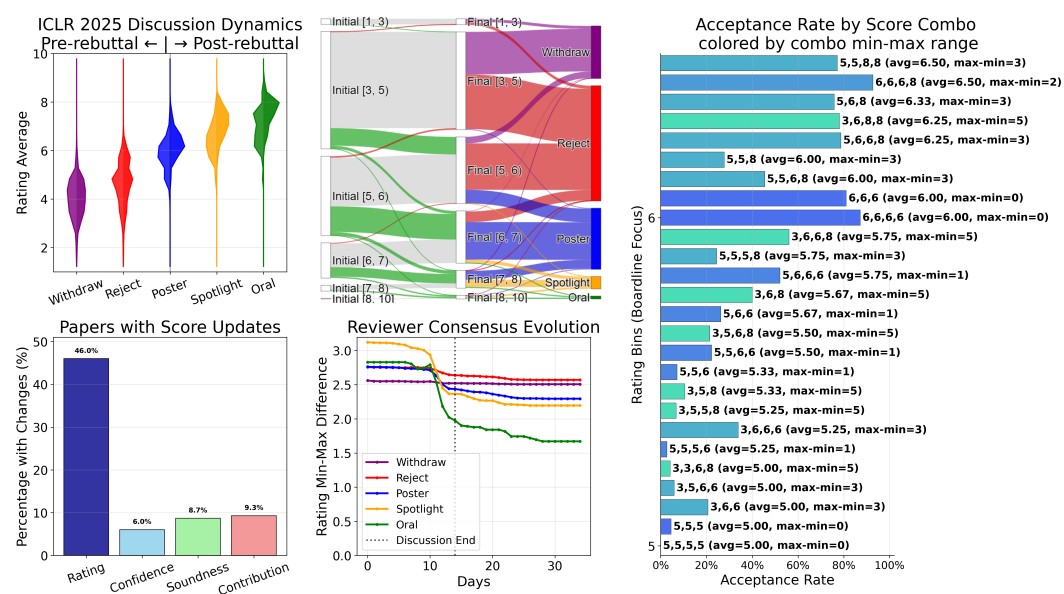

Figure 4: **Rebuttal dynamics in ICLR 2024**. (top-left) Ridge plots of pre- vs. post-rebuttal ratings across final decision tiers. (top-mid) Sankey plot tracking score trajectories from initial rating bins to final bins and final statuses. (lower-left) Distribution of score updates across review dimensions. (lower-mid) Reviewer consensus evolution measured by min–max rating difference. (right) Acceptance rates conditioned on reviewer score combinations with frequency cutoff=30.

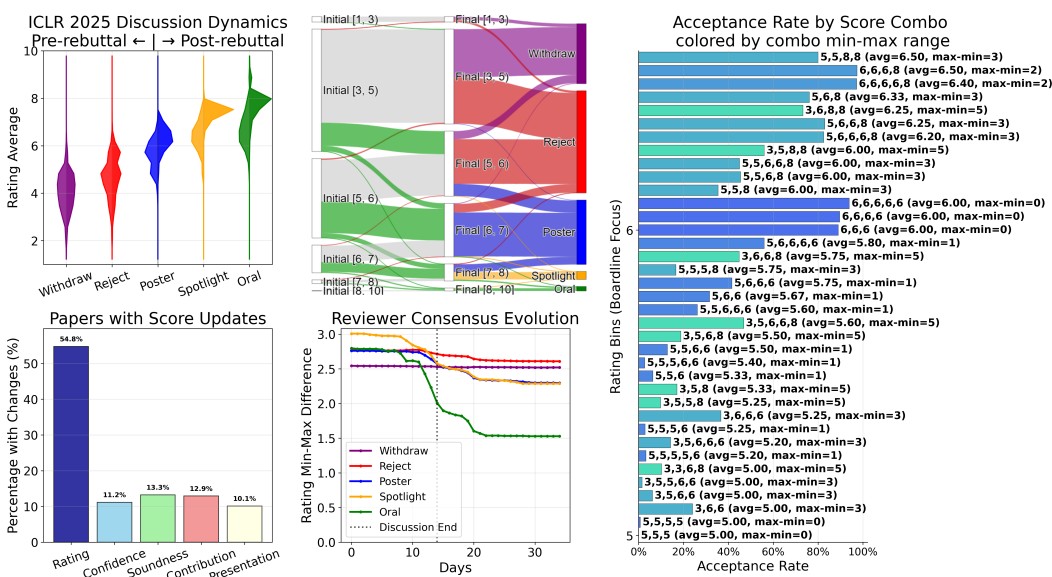

Figure 5: **Rebuttal dynamics in ICLR 2025**. (top-left) Ridge plots of pre- vs. post-rebuttal ratings across final decision tiers. (top-mid) Sankey plot tracking score trajectories from initial rating bins to final bins and final statuses. (lower-left) Distribution of score updates across review dimensions. (lower-mid) Reviewer consensus evolution measured by min–max rating difference. (right) Acceptance rates conditioned on reviewer score combinations with frequency cutoff=30.

## C  AUTHOR–AFFILIATION METADATA ACQUISITION

In this section, we describe our methodology for extracting structured author–affiliation–department–country metadata at scale and introduce evaluation metrics to assess extraction quality. Constructing such a dataset is traditionally infeasible: most conferences do not release detailed author–affiliation metadata, and due to diverse paper formats and complex author–affiliation mappings—especially in the presence of multiple affiliations—manual extraction is both time-consuming and expensive.

Conventional institutional rankings (e.g., CSRankings, U.S. News) rely on opaque, top-down methodologies that are difficult to audit or reproduce. In contrast, large language models (LLMs) provide a scalable and cost-efficient alternative for parsing and structuring paper metadata. We benchmark the ability of modern LLMs to automatically extract author–affiliation associations using instruction prompting and produce consistent, verifiable metadata.

**Task Formulation**   Formally, given a set of input documents $\mathcal{D} = \{d^1, d^2, \dots\}$, the task is to predict a structured set of metadata tuples:

$$\mathcal{T}_\mathcal{D} = \{(a^i, \mathcal{A}^i, e^i)\}_{i=1}^{|\mathcal{D}|}$$

where $a^i = \{a_1^i, a_2^i, \dots\}$ denotes the authors in $i$-th document, $\mathcal{A}^i = \{\{a_{1\alpha}^i, a_{1\beta}^i, \dots\}, \{a_{2\alpha}^i, a_{2\beta}^i, \dots\}, \dots\}$ is the set of affiliations associated with $a^i$, and $e^i$ is the corresponding email address. Each affiliation $a_{jk}^i$ is further mapped to a country $c_{jk}^i$.

**Evaluation Metrics**   To assess the quality of metadata extraction, we define a set of metrics that measure structural consistency between predicted and expected fields in each document. In particular, we evaluate whether the number of predicted affiliations and emails matches the number of authors.

We introduce the mismatch indicator function:

$$\mathbf{1}(x, y) = \begin{cases} 1 & \text{if } |x| \neq |y| \\ 0 & \text{otherwise} \end{cases}$$

which returns 1 if the two arguments differ in cardinality and 0 otherwise. This operator captures structural inconsistencies across fields. Given the extracted tuple set $\mathcal{T}^i = (a^i, \mathcal{A}^i, e^i)$ for a document, we define the affiliation mismatch $\delta_{\text{aff}}^i = \mathbf{1}(\mathcal{A}^i, a^i)$ and email mismatch $\delta_{\text{email}}^i = \mathbf{1}(e^i, a^i)$, where $\mathcal{A}^i$, $a^i$, and $e^i$ denote the sets of predicted affiliations, authors, and emails.

In addition to structural mismatches, we define a parsing failure indicator $\delta_{\text{parse}}^i$, which is set to 1 if the LLM output for document $d$ cannot be parsed into the expected structured format (e.g., malformed delimiters, inconsistent field lengths, or invalid JSON), and 0 otherwise.

The overall success rate across the corpus $\mathcal{D}$ is then defined as:

$$\text{Success Rate} = 1 - \frac{1}{|\mathcal{D}|} \sum_{i=0}^{|\mathcal{D}|} \left( \delta_{\text{aff}}^i \vee \delta_{\text{email}}^i \vee \delta_{\text{parse}}^i \right)$$

This metric captures the fraction of documents for which both affiliation and email counts are structurally aligned with the author list, which is a necessary prerequisite for reliable metadata extraction.

**Evaluation**   Given the scale of this benchmark, where token consumption can easily exceed the billion-token level ($10^9$), we employ a suite of efficient, batchable, and lower-cost pretrained LLMs that also achieve performance comparable to state-of-the-art proprietary models. Specifically, we evaluate four pretrained models from the GLM family: `glm-4`, `glm-4-plus`, `glm-4-air`, `glm-4-flash`, and `glm-3-turbo`. These models offer a strong trade-off between accuracy and throughput, making them well-suited for large-scale structured extraction tasks.

We conduct the evaluation across seven major machine learning venues over the past five years, covering more than 70,000 papers, with each experiment processing approximately 150 million tokens. Results are summarized in Table 1, highlighting structural consistency in extracting author–affiliation–email triplets. Token usage and associated inference costs are detailed in Table 2, Token Consumption Section. We omit benchmarking `glm-4` due to its prohibitive cost—each trial with 100 million tokens exceeds $1,000.

Table 1: **Evaluation of LLM extraction accuracy on author–affiliation–email triplet consistency.** We report the structural mismatch rates for affiliations and emails, the unparseable output rate ($\delta_{\text{parse}}$), and the overall success rate across five GLM family models evaluated on the metadata benchmark.

| Model | $\delta_{\text{aff}}$ (%) | $\delta_{\text{email}}$ (%) | $\delta_{\text{parse}}$ (%) | Success Rate (%) |
|---|---|---|---|---|
| glm-4-plus | 5.01% | 4.94% | 0.81% | 86.82% |
| glm-4-air | 49.98% | 17.11% | 0.51% | 44.73% |
| glm-4-flash | 76.39% | 43.27% | 0.62% | 18.52% |
| glm-3-turbo | 76.07% | 32.34% | 1.34% | 20.90% |

Table 2: **Cross-venue token usage and affiliation disclosure (2021–2025).** Token consumption statistics and available structured affiliation metadata across major AI/ML venues.

| Venue Metadata | | | Affiliation Disclosure | | | Token Consumption | | |
|---|---|---|---|---|---|---|---|---|
| Venue | Years | #Papers | Author–Affil. | Affil.–Country | % Multi-Affil. | Prompt Tokens | Completion Tokens | Total Tokens |
| AAAI | 2021–2025 | 11347 | Partial | No | 38.10% | 1.80M | 19.22M | 21.02M |
| ACL | 2021–2024 | 5724 | No | No | 39.47% | 0.88M | 8.86M | 9.74M |
| CVPR | 2021–2024 | 8803 | No | No | 38.12% | 1.36M | 12.68M | 14.03M |
| EMNLP | 2021–2024 | 7365 | No | No | 32.94% | 1.12M | 11.35M | 12.47M |
| ICCV | 2021,2023 | 3768 | No | No | 34.95% | 0.57M | 5.44M | 6.01M |
| ICML | 2023–2024 | 4438 | No | No | 39.12% | 0.63M | 7.00M | 7.62M |
| IJCAI | 2021–2024 | 3484 | No | No | 34.67% | 0.53M | 5.75M | 6.28M |
| NeurIPS | 2021–2024 | 14190 | No | No | 31.28% | 2.09M | 17.35M | 19.44M |

*Note:* "Author–Affiliation" refers to structured author–affiliation pair disclosure in submission metadata. "Partial" indicates that only one affiliation per author is disclosed, even when multiple affiliations are listed in the camera-ready version. No venues disclose structured affiliation–country information. Multi-affiliation percentages are estimated from parsed camera-ready PDFs using our automated extraction pipeline enabled by *glm-4-plus*.