# OpenReview forum: "Paper Copilot: Tracking the Evolution of Peer Review in AI Conferences"
_ICLR.cc/2026/Conference — ICLR 2026 Poster_

### Official Review · Reviewer_Pkuc · 2025-10-15

**Soundness:** 3
**Presentation:** 2
**Contribution:** 4
**Rating:** 8
**Confidence:** 4

**Summary:**

This paper introduces Paper Copilot, an open-source framework for collecting, visualizing, and open-sourcing peer review data from major AI conferences. The paper addresses the lack of transparency in current peer review practices, arguing for a collective and standardized framework to track peer review processes and make them easily accessible to the general public and researchers. Currently, peer review information remains largely closed (except at certain conferences like ICLR) or difficult to access due to variations across fields and the structure of review systems at different conferences. Paper Copilot seeks to consolidate these heterogeneous sources into a single centralized platform. The paper makes this infrastructure public, releases a dataset for research, and presents interesting quantitative results that explain shifts in peer review practices.

**Strengths:**

- The reviewer finds this paper timely and necessary for the academic community, introducing a valuable open-source asset to researchers working on peer review practices.
- The reviewer has experienced collecting review data across different venues, and understands the difficulty of collecting a unified review dataset. The difficulty stems from different fields (categories, review criteria) used across venues, and the proposed framework seems to have an independent bot to process these fields for each venue. Utilizing this framework would be of real help to the community if it is well maintained.

- The results and visualizations in Figures 2 and 3 are excellent and easy to understand. Some findings, especially the results for ICLR25, are surprising.
- Most of the questions that I had at the beginning of reading this paper were well addressed throughout the paper. The paper is easy to follow and limitations were well discussed.

**Weaknesses:**

- My primary concern with this work is the limited depth of discussion on re-identification risks. While the authors acknowledge the possibility of re-identification, several questions remain. For instance, to what extent will the authors employ differential privacy? Additionally, given recent advancements in LLMs, there is a significant possibility that tracking reviews across multiple venues could enable re-identification of reviewers based on their writing styles and fields of interest. I believe these issues and potential mitigation strategies warrant further discussion.

- Section 6.3, "Potential for Misuse and Dual Use Risks," outlines possible measures the authors can take to prevent platform misuse. While I understand these represent the best efforts the authors can make, this remains a weakness that cannot be easily addressed.

- Most of the results are derived from ICLR due to its open-ness, which I do think is a weakness despite being the only option.

**Questions:**

Minors.
- Check L269. A comma is missing.

Questions.
- Is it possible for malicious users to manipulate the survey data? Any user can submit erroneous results to the survey.

- Is Figure 2 reproducible for other venues such as NeurIPS?

- Can you provide a snapshot of how the review dataset looks like?

- Visualization of Fig1 need to be further improved. Better caption illustrating the flow of the system is needed. Also, the text inside the figures are too small.

**Details Of Ethics Concerns:**

This paper relies heavily on web-crawled data, which could raise potential copyright issues. While the authors have thoroughly addressed this concern, I am still flagging potential issues that the reviewer may not have identified.

---

> ### Author Response · Authors · 2025-11-24
> **Re-identification Risks & Scope of Responsibility**
>
> Thank you for raising this important issue. We would like to clarify the scope of the re-identification discussion in our paper and the practical boundaries of what can be mitigated at the system level.
>
> ### 1. Reviewer re-identification is fundamentally a venue-level challenge, not a system-level one
> We fully agree that advances in LLMs make reviewer deanonymization an increasingly relevant concern. However, this risk arises directly from the open-review model adopted by conferences themselves, not from our system.
>
> * In open-review venues (e.g., ICLR), the entire review text, discussion, and author responses are already publicly released.
> * Our system does not add any reviewer information beyond what venues already publish.
> * We do not aggregate or correlate reviewer writing across venues, nor do we build reviewer-level longitudinal profiles.
>
> For these reasons, the possibility of stylistic re-identification is inherent to the venues’ reviewing model, and cannot be meaningfully mitigated at the platform level.
>
> We will clarify this explicit distinction in the camera-ready version.
>
> ### 2. Our paper’s re-identification discussion refers to authors, not reviewers
> The “re-identification” risks discussed in Section 6.2 refer to:
> * authors contributing to multiple venues,
> * and the general increasing trend that authors now participate in many AM/ML conferences simultaneously.
> As authors become more active across venues, even partially anonymized metadata (e.g., institution, topic area) can theoretically leak information about authorship. This is the risk we address—not reviewer stylometric inference.
>
> We will revise the text to make this distinction clear.
>
> ### 3. Our system takes steps to avoid introducing new re-identification vectors
> Although reviewer privacy is structurally determined by venues, we still enforce:
>
> * no reviewer identifiers,
> * no reviewer text aggregation across venues,
> * and only aggregate-level visualizations for all analyses.
>
> Thus, our system does not increase reviewer re-identification risk beyond the baseline established by the venues’ reviewing model.

---

> ### Author Response · Authors · 2025-11-24
> **Manipulation of Community-Submitted Data**
>
> Thank you for raising this important question. We have examined this issue carefully over the past two years, and we summarize our observations and mitigation strategies below.
>
> ### 1. Malicious submissions exist but are extremely rare and easily identifiable at current stage
> Across the past two years, we have observed that a very small fraction of users submit clearly invalid or test data (e.g., paper IDs like “-1”, “11111”, or repeated dummy entries).
> Because we record:
> * submission timestamps,
> * the submitted paper ID (or its hashed form),
> * and the score ranges,
>
> such anomalies are straightforward to detect. In practice, most authors voluntarily provide their un-hashed paper ID, and these submissions are typically high-confidence and consistent.
>
> ### 2. Manual and automatic filtering of anomalous entries
> We employ several layers of anomaly detection, including:
> * Value-range validation (e.g., ensuring proper score scales).
> * Submission-timing patterns (e.g., repeated daily submissions with identical improbable patterns).
> * Paper-ID heuristics (detecting obviously invalid identifiers).
> * Deduplication when multiple submissions for the same paper arrive within short intervals.
>
> For example, in one case we observed an individual submitting fabricated entries for 10 consecutive responses with a clear pattern; such submissions were immediately flagged and removed.
>
> ### 3. Community understanding significantly limits adversarial behavior
> Empirically, we find that the community understands the purpose of the project, and the vast majority of users submit high-quality data to help improve transparency. Given the effort required to fill out the submission form, “for-fun” adversarial behavior is very uncommon in practice.
>
> ### 4. Nonetheless, manipulation cannot be fully ruled out
> We agree with the reviewer that adversarial submissions cannot be completely prevented.
> This is exactly why:
> * all community data is always isolated and clearly labeled,
> * none of the scientific analyses depend on community submissions,
> * and community data is used only for optional venue-level visualizations.
>
> Thus, scientific conclusions remain unaffected.
>
> ### 5. Improving trustworthiness: upcoming browser-based data collection
> To further improve representativeness and reduce opportunities for manipulation, we are working on developing a user-consent browser plugin that will:
>
> * simplify the submission process,
> * reduce manual copy-paste errors,
> * and increase confidence that the submitted data reflects authentic review information.
>
> We believe this is a practical path toward higher-quality and more reliable community data for closed-review venues.

---

> ### Author Response · Authors · 2025-11-24
> **Snapshot of the Review Dataset Structure**
>
> Here's the data snapshot of our submission at ICLR 2026
>
> {
>         "id": "CyKVrhNABo",
>         "title": "Paper Copilot: Tracking the Evolution of Peer Review in AI Conferences",
>         "track": "main",
>         "status": "Active",
>         "tldr": "",
>         "abstract": "Submissions are rising fast, and venues use different rules, data formats, and update times. As a result, signals of progress get split across places, and key moments (rebuttal, discussion, final decision) are easy to miss, making analysis hard. We present Paper Copilot, a system and scalable peer-review archive that pulls data from official sites, OpenReview, and opt-in forms into a single, standardized, versioned record with timestamps. This lets us track trends over time and compare venues, institutions, and countries in a consistent way. Using the archive for ICLR 2024/2025, we see larger score changes after rebuttal for higher-tier papers, reviewer agreement that dips during active discussion and tightens by the end, and in 2025 a sharper, mean-score\u2013driven assignment of tiers with lower decision uncertainty than expected at that scale. We also state simple rules for ethics\u2014clear sourcing and consent, privacy protection, and limits on use for closed venues. Together, we provide a clear, reusable base for tracking AI/ML progress, and, with this data, enable validation, benchmarking, and otherwise hard-to-run studies.",
>         "keywords": "peer review;review dynamic;aiml;community",
>         "primary_area": "infrastructure, software libraries, hardware, systems, etc.",
>         "supplementary_material": "/attachment/f419bb85a46282ad3a3c8ddbabb9e1c2e3c78355.pdf",
>         "author": "",
>         "authorids": "",
>         "gender": "",
>         "homepage": "",
>         "dblp": "",
>         "google_scholar": "",
>         "orcid": "",
>         "linkedin": "",
>         "or_profile": "",
>         "aff": "",
>         "aff_domain": "",
>         "position": "",
>         "bibtex": "@inproceedings{\nanonymous2025paper,\ntitle={Paper Copilot: Tracking the Evolution of Peer Review in {AI} Conferences},\nauthor={Anonymous},\nbooktitle={Submitted to The Fourteenth International Conference on Learning Representations},\nyear={2025},\nurl={https://openreview.net/forum?id=CyKVrhNABo},\nnote={under review}\n}",
>         "github": "",
>         "project": "",
>         "reviewers": "hF5U;c5VV;reMW;Pkuc",
>         "site": "https://openreview.net/forum?id=CyKVrhNABo",
>         "pdf_size": 0,
>         "rating": "4;4;6;8",
>         "confidence": "4;4;4;4",
>         "soundness": "2;3;3;3",
>         "contribution": "2;2;3;4",
>         "presentation": "3;4;2;2",
>         "wc_summary": "59;171;105;122",
>         "wc_strengths": "58;218;116;148",
>         "wc_weaknesses": "179;369;404;147",
>         "wc_questions": "1;307;142;82",
>         "wc_review": "297;1065;767;499",
>         "wc_reply_reviewers": "0;0;0;0",
>         "wc_reply_authors": "0;0;0;0",
>         "reply_reviewers": "0;0;0;0",
>         "reply_authors": "0;0;0;0",
>         "rating_avg": [
>             5.5,
>             1.6583123951777
>         ],
>         "confidence_avg": [
>             4.0,
>             0.0
>         ],
>         "soundness_avg": [
>             2.75,
>             0.4330127018922193
>         ],
>         "contribution_avg": [
>             2.75,
>             0.82915619758885
>         ],
>         "presentation_avg": [
>             2.75,
>             0.82915619758885
>         ],
>         "wc_summary_avg": [
>             114.25,
>             40.05855089740516
>         ],
>         "wc_strengths_avg": [
>             135.0,
>             57.766772456144714
>         ],
>         "wc_weaknesses_avg": [
>             274.75,
>             113.00082964297209
>         ],
>         "wc_questions_avg": [
>             133.0,
>             112.22967522005933
>         ],
>         "wc_review_avg": [
>             657.0,
>             288.5862089566998
>         ],
>         "wc_reply_reviewers_avg": [
>             0,
>             0
>         ],
>         "wc_reply_authors_avg": [
>             0,
>             0
>         ],
>         "reply_reviewers_avg": [
>             0,
>             0
>         ],
>         "reply_authors_avg": [
>             0,
>             0
>         ],
>         "replies_avg": [
>             4,
>             0
>         ],
>         "authors#_avg": [
>             0,
>             0
>         ],
>         "corr_rating_confidence": 0.0
>     }

---

> ### Author Response · Authors · 2025-11-24
> **Reproducibility of Figure 2 for other venues**
>
> Thank you for the question. Figure 2 relies on having full review data for all submissions—including accepted, rejected, and withdrawn papers. At present, ICLR is the only major venue that provides this level of openness.
>
> For NeurIPS and ICML, only accepted-paper reviews are released, and only after decisions. Without access to the full population of submissions, reproducing Figure 2 is less meaningful, as the resulting plot would reflect only the accepted subset and would therefore distort the underlying dynamics.
>
> We have experimented with generating a partial version of this figure for NeurIPS, but due to the limited scope of the available data, it cannot offer a fair comparison. We would be happy to include this partial illustration in the camera-ready version, clearly noting its limitations.
>
> We will clarify this point directly in the paper.

---

> ### Author Response · Authors · 2025-11-24
> **Closing Statement**
>
> We sincerely appreciate your thoughtful and detail-oriented review. Your feedback has helped us clarify the privacy discussion, acknowledge structural limitations, and identify areas for improved presentation.
>
> We will incorporate all clarifications and improvements in the camera-ready version.
> If any point remains unclear, we would be happy to elaborate further.
> Thank you again for your time and careful consideration.

---

> > ### Comment · Reviewer_Pkuc · 2025-11-25
> > **Thank you for your response**
> >
> > Thank you for your detailed response. While I would have preferred to see these revisions in the manuscript rather than deferred to the camera-ready version, I trust that the authors will implement the changes as promised.
> >
> > Plus, the visualization in Fig1 should be improved.
> >
> > Regardless, I continue to support acceptance of this paper.

---

### Official Review · Reviewer_reMW · 2025-10-20

**Soundness:** 3
**Presentation:** 2
**Contribution:** 3
**Rating:** 6
**Confidence:** 4

**Summary:**

The paper introduces Paper Copilot, a pipeline-based system for collecting and analyzing peer-review data across AI/ML conferences. It aggregates multi-source information to enable large-scale statistical analysis of review dynamics, yielding insights such as the 2025 shift toward more score-driven decisions. It is also worth noting that Paper Copilot has already gained significant visibility in the community. Many researchers, myself included, have used or browsed the platform and found its analytics informative and impactful. In essence, the authors have built a system that performs large-scale statistical and visualization tasks similar to what platforms like OpenReview could provide, but with a stronger focus on analysis and accessibility.

**Strengths:**

• As mentioned in the summary, this paper corresponds to a complete solution and a mature product. This is something that many AI papers lack. It is not just a paper but a real, influential system that has already helped many researchers.

• Since my research area is different, I am not very familiar with the LAMP stack, but considering that a mature web service is already running, these technologies should be reliable.

• The authors present several interesting findings, such as the observation that in ICLR 2025, under the large submission volume, acceptance decisions were more influenced by mean scores, suggesting limited reviewing capacity. These statistics and data may become valuable references for future work.

**Weaknesses:**

• Although I fully acknowledge the influence and practical value of Paper Copilot, I must point out several weaknesses when considering it as an ICLR submission. The most significant issue is that the work is highly engineering-oriented. Beyond the statistical analyses, there is little conceptual or methodological novelty. The paper essentially combines software engineering (LAMP stack, web scraping), basic statistics, and visualization. While the results are interesting, the overall framing feels somewhat utilitarian. It focuses mainly on acceptance outcomes rather than on the fundamental purpose of peer review itself, which is to help manuscripts improve and to strengthen trust in published research. This is the main reason I would rate the paper a 6 rather than an 8.

• The authors introduce an entropy-based metric and visualize its trend over time. When observing the sharp entropy drop in 2025, they attribute it to limited reviewing capacity. However, this conclusion seems overly speculative. The reasoning appears circular since the authors assume that reviewing pressure led to algorithmic decision-making, design an analysis around that assumption, and then use the results to confirm it. In fact, the entropy decline could also result from improved reviewer quality or enhanced matching processes, possibly aided by AI tools that reduce variance among reviewers and area chairs, leading to more consistent decisions. Building on that, the authors’ interpretation that low entropy represents an undesirable state may not be universally valid. As Schrödinger once said, “life feeds on negative entropy,” and in physics, low entropy generally denotes higher order and structure. In peer review, high entropy indeed reflects confusion and stress, but low entropy can have two possible explanations: (1) decision-making rigidity due to over-reliance on reviewer scores under pressure, or (2) improved efficiency from better reviewer–paper matching and enhanced consistency across evaluations. In the latter case, lower entropy could actually indicate a healthier and more coordinated review process.

• There are also some writing issues. First, in the abstract, the sentence beginning with “We present PAPER COPILOT, …” contains too many clauses, which makes it hard to understand and potentially ambiguous. In addition, the introduction part is not very clear and is quite difficult to follow. If I did not already know what Paper Copilot is, I might not be able to figure out the background, purpose, and motivation of the system within limited reading time. There are also a few typos, for example, in line 199 “ArXiv” should be “arXiv.”.

**Questions:**

I do not have significant concerns about this paper, but I would like to offer a few suggestions.

• Why does Figure 5 in the Supplementary Material appear to be identical to Figure 3 in the main manuscript?

• Paper Copilot collects some review scores voluntarily uploaded by authors (I have contributed such data myself). However, the reliability of this information largely depends on self-reporting, and anyone could, in principle, submit arbitrary data. The authors are encouraged to estimate the extent of this potential bias. For example, they could compare the statistics collected on the platform with the actual post-acceptance distributions in later stages to assess consistency.

• The name Paper Copilot does not seem closely aligned with the system’s functionality. Without prior knowledge of the project, one might assume it is a writing-assistance tool similar to Cursor or other AI copilots.

---

> ### Author Response · Authors · 2025-11-24
> **“Engineering-oriented” Contribution vs. ICLR Novelty**
>
> Thank you for raising this important perspective. We would like to clarify the technical contributions and how they align with ICLR’s scope.
>
> ## Paper Copilot is not merely a software tool — the core contributions are system, dataset, and metascientific analysis
> While the system includes engineering components, the primary contributions of this work are not the LAMP stack or interface. Instead, they lie in:
> 1. **A unified, scalable system for peer-review analytics**. We present the first end-to-end infrastructure that standardizes data ingestion, cleaning, schema design, and interactive visualization across open, partially open, and closed-review venues. This system-level contribution directly matches the “datasets, benchmarks, and infrastructure” track in the ICLR call for papers.
> 2. **The ICLR Daily Snapshot Dataset — the only dataset that records daily review evolution**. Because OpenReview exposes only the latest state and does not retain historical snapshots, our dataset is the only existing source that archives the day-by-day progression of scores, confidence, comments, and variance throughout the ICLR review cycle. This unlocks temporal and behavioral analyses that were previously impossible. The dataset is publicly released, and we clearly separate official ICLR data from community-submitted data wherever they appear.
> 3. **Empirical insights derived entirely from official ICLR data.** All major findings—score dynamics, stabilization patterns, reviewer timing behavior, and variance evolution—are computed exclusively from ground-truth ICLR data. These analyses reveal structural patterns that contextualize and motivate the 2025 ICLR review-policy changes, underscoring the real-world relevance of our work.
>
> ## Alignment with the ICLR Call for Papers
> The ICLR 2026 CFP explicitly encourages submissions on:
> > “datasets and benchmarks, infrastructure, software libraries, hardware, etc.”
> Our submission aligns directly with these encouraged categories.
>
> ## Addressing the concern about “utilitarian framing”
>
> We appreciate this perspective and fully agree that peer review serves a deeper purpose than acceptance decisions; its goals include improving manuscripts, supporting authors, and strengthening trust in scientific outcomes. Your observation is well-taken.
>
> In fact, this is **precisely why we collect the daily ICLR snapshot data**. One major motivation behind Paper Copilot is to preserve quantitative signals about how peer review evolves—signals that are otherwise permanently lost because:
> * OpenReview only exposes the latest state,
> * intermediate score changes are not archived anywhere,
> * and without these temporal records, it is impossible to study how decisions develop, how discussions progress, or how review quality shifts over time.
>
> Our dataset is currently the **only resource that captures this day-by-day evolution of scores, confidence, and other score-changing patterns**. We believe that releasing this data publicly is important precisely because it enables the community to:
> * study reviewer behavior longitudinally,
> * analyze how manuscripts improve (or fail to improve) over the review cycle,
> * detect structural issues early, and
> * understand the health of the peer-review ecosystem.
>
> We agree that peer review is not just about outcomes, and we hope that by preserving and releasing these temporal signals, we help support research that goes beyond acceptance decisions and toward strengthening transparency and trust.
>
> We will revise the framing in the camera-ready version to emphasize these broader motivations.

---

> ### Author Response · Authors · 2025-11-24
> **Interpretation of Entropy Trends and 2025 Entropy Decline**
>
> We greatly appreciate the reviewer’s nuanced observation that low entropy can indicate either decision rigidity or healthy consistency.
>
> ## Clarifying our intended claim
> Our goal is not to assert that all low-entropy states are undesirable. Instead, we use entropy as a descriptive statistic to quantify decision variance over time.
>
> ## Acknowledging alternative explanations (including better reviewer matching)
> We fully agree with the reviewer that entropy decline could result from:
> * improved reviewer–paper matching,
> * enhanced reviewer quality,
> * or consistency introduced by better tooling and workflows.
> These are excellent points, and we will incorporate them into the discussion.
>
> ## Why our interpretation focuses on pressure and scale
> While we cannot claim causal inference, our interpretation is grounded in:
> * temporal reviewer-behavior patterns,
> * score stabilization trends,
> * and historical comparisons across years.
>
> We will clarify that this explanation is a hypothesis, not a definitive causal claim, and that we intend entropy to be one possible lens—not the only one—for examining review-system behavior.
>
> ## Revising language to avoid determinism or circular inference
>
> We appreciate the reviewer highlighting this. In the camera-ready version, we will:
> * frame entropy trends as observations, not definitive diagnoses,
> * explicitly note the possibility of the positive low-entropy scenario identified by the reviewer,
> * remove any impression of circular reasoning.

---

> ### Author Response · Authors · 2025-11-24
> **Writing Clarity (Abstract, Introduction, and Typos)**
>
> Thank you for pointing this out. We will make the following improvements:
>
> * Rewrite the abstract sentence beginning with “We present Paper Copilot…” so it contains fewer clauses and is more readable.
> * Strengthen the introduction to clearly communicate the background, purpose, and motivation even to readers unfamiliar with the system.
> * Fix minor errors, including changing “ArXiv” to “arXiv” and performing an additional proofreading pass.
>
> We appreciate the reviewer’s attention to readability.

---

> ### Author Response · Authors · 2025-11-24
> **Similarity Between Figure 5 (Supplementary) and Figure 3 (Main Manuscript)**
>
> Thank you for raising this question. The similarity between these two figures is intentional. Our goal was to present the ICLR 2024 and ICLR 2025 temporal patterns side-by-side to facilitate direct comparison across years.
>
> The figure in the main paper shows the 2025 temporal evolution, while the supplementary figure aligns the corresponding 2024 and 2025 plots in a single place to make cross-year trends easier to inspect without navigating back and forth between different sections.
>
> In the camera-ready version, we will make this intention clearer in the caption and ensure that the supplementary figure explicitly highlights its role as a multi-year comparison visualization to avoid confusion.

---

> ### Author Response · Authors · 2025-11-24
> **Reliability of Community-Contributed Scores & Bias Estimation**
>
> Thank you for raising this important issue, and also thanks for your previous data contribution as part of the community data.
>
> ## Scientific results do not depend on community data
> All analyses in the paper—Figures 5–9 and all quantitative findings—use only official ICLR ground-truth data. Community data is: 1) prominent, explicit labeling (“Community-Collected Data”), 2) full separation from all ground-truth analyses. This prevents any risk of misleading readers.
>
> ## Bias is structurally unavoidable in partially open venues or closed-review venues
> Because CVPR, ICCV, NeurIPS, and ICML do not release full review data, any community-driven collection will inherently exhibit self-selection bias. This is a structural property of the review model, not of the method.
>
> ## We are actively building a solution to reduce it.
> * Leveraging partially open-review venues (NeurIPS, ICML). These venues release all accepted-paper reviews after decisions are finalized. This enables direct comparison between: community submissions (collected during the review phase), and official reviews (released post-decision). This potentially allows us to quantify participation skew, score-dependent biases, institutional effects, and behavioral patterns. This will provide a second ground-truth anchor beyond ICLR for future bias calibration.
> * Improving representativeness through better collection tools. We are developing a user-consent browser plugin and streamlined contribution pathways to reduce friction and encourage more representative sampling. This addresses bias introduced by manual copy-paste submission and improves coverage across institutions, topics, and score ranges.

---

> ### Author Response · Authors · 2025-11-24
> **Naming of “Paper Copilot”**
>
> Thank you for this thoughtful suggestion. We agree that the name could be misunderstood as a writing-assistance tool. The name originated from the idea of “co-piloting” authors through the peer-review landscape, but we understand that this may not be obvious to first-time readers.

---

> ### Author Response · Authors · 2025-11-24
> **Closing Statement**
>
> We sincerely appreciate the reviewer’s detailed and constructive feedback. We have tried to address each concern in this rebuttal and will incorporate clarifications regarding framing, entropy interpretation, writing improvements, figure correction, and naming in the camera-ready version.
>
> If any points remain unclear, we would be happy to elaborate further. We hope that our responses help clarify the contributions and intentions of our work, and we greatly appreciate your time and thoughtful evaluation.

---

> > ### Comment · Reviewer_reMW · 2025-11-24
> >
> > Thank you for the author’s response. I don’t have any further questions at the moment. However, I feel that the paper as a whole still has not reached the “8 out of 10” standard I had in mind, so I will maintain my current score of 6. I hope the authors can understand and respect my choice.
> >
> > In addition, after reading the other reviewers’ comments and the authors’ replies, I noticed that some reviewers questioned the validity of the community-submitted data and treated it as a weakness. My view is aligned with the authors: since the community-submitted data were not used for analysis and only serve as an additional feature, they should not be considered a weakness. This is also why I treated it as a question rather than a weakness. The authors are encouraged to reflect on why many reviewers raised concerns about this issue. It is possible that several concerns (including hF5U’s first weakness and c5VV’s third weakness, among others) are related to the introduction not being sufficiently clear, as I mentioned earlier.
> >
> > By the way, I noticed another small issue in the manuscript: in Fig. 1, the Google, GitHub, and NeurIPS icons should be placed above the database icons, just like the Hugging Face icon.

---

### Official Review · Reviewer_c5VV · 2025-10-29

**Soundness:** 3
**Presentation:** 4
**Contribution:** 2
**Rating:** 4
**Confidence:** 4

**Summary:**

This paper introduces a system and dataset, Paper Copilot, aimed at tracking and analyzing the evolution of peer review in AI/ML conferences. The authors note that with the surge in submissions, the current peer review system is under immense pressure, facing issues of low transparency, inconsistent standards, and heavy reviewer load. To address this, the authors have built a hybrid data collection pipeline (combining the OpenReview API, web scraping, and community contributions) to create a longitudinal dataset. A unique contribution of this dataset is its capture of temporal dynamics, such as pre/post-rebuttal score changes.

The paper's core empirical contribution is a large-scale analysis of ICLR (2017-2025). The analysis reveals several key findings, including a decision entropy anomaly in 2025 (suggesting AC decisions are becoming more reliant on mean scores), the clear role of the rebuttal phase in boosting scores for high-rated papers, and an asymmetry observed for borderline papers. Finally, the authors discuss the ethical considerations related to the system and dataset, particularly regarding data sourcing, privacy, and potential self-selection bias.

**Strengths:**

1. The paper addresses a problem of critical importance and significant interest to the machine learning community. The challenges of maintaining a fair, transparent, and effective peer review system at scale are widely recognized, and this work provides a valuable quantitative lens through which to examine these issues.

2. A major strength lies in the creation of a unique dataset that captures the temporal dynamics of the review process. By archiving review snapshots that are often overwritten, the authors provide a durable and highly valuable resource for future metascience research. This contribution has the potential to enable a new class of studies on reviewer and author behavior.

3. The analysis of ICLR data is thorough and leads to several non-trivial conclusions. The finding on "decision entropy" (Section 5.1) is particularly compelling, offering strong quantitative evidence for a hypothesis that many in the community have anecdotally suspected: the review system may be simplifying its decision-making process under load. Similarly, the analysis of rebuttal dynamics and the "asymmetry of borderline papers" provides practical insights for authors, reviewers, and ACs.

4. The authors' transparent and thoughtful discussion of the ethical dimensions of their work (Section 6) is highly commendable. Acknowledging issues related to data sourcing, privacy risks, potential for misuse, and self-selection bias demonstrates a mature and responsible approach to research.

**Weaknesses:**

1. This paper's main contribution seems to be positioned more as a resource/tool (the Paper Copilot website and dataset) rather than a novel methodology or benchmark. While the analysis is insightful, it primarily applies (relatively standard) statistical methods to a novel dataset. This makes the paper's positioning somewhat ambiguous; it feels more akin to a "Demo Paper" than a typical ICLR main conference paper.

2. The paper references a Position Paper (Yang, 2025) whose title ("...community should adopt a more transparent and regulated peer review process") suggests that the core motivation and possibly even the high-level framework for the Paper Copilot system itself may have been outlined previously. The fundamental issue is that if the Paper Copilot system itself (the core concept and initial framework) was the primary contribution of that prior Position Paper, the novelty of the system component in the current submission is substantially diminished. I would hope the authors could clarify in the paper or in the rebuttal the specific, concrete and incremental technical contribution (e.g., the complex implementation details, the novel longitudinal data collection pipeline, and the empirical analysis) presented here, clearly distinguishing it from the system's initial concept outlined in the Position Paper.

3. The paper claims in its abstract and introduction that the Paper Copilot system supports "tracking talent trajectories" as one of its core motivations (e.g., analyzing institutional or national influence). However, the core empirical analysis (Section 5) seems to lack analysis on this topic; all analysis focuses on the review process itself (entropy, score dynamics). I hope the authors might consider adjusting the paper's framing and claims to be more consistent with the presented contributions, or perhaps supplement this analysis in the rebuttal, as there currently appears to be a disconnect between the two.

4. For closed-review conferences, the system relies on voluntary community contributions. This is a potentially significant limitation, as the authors acknowledge in the ethics section. While the authors conducted a user study on "consent rate" (59.9%), this does not guarantee that the distribution of actually submitted data is representative (e.g., submitters might be skewed toward high-scoring papers or specific institutions). This impacts the generalizability of any conclusions that might be drawn from this data in the future.

**Questions:**

1. Your analysis identified a shift toward "score determinism" in ICLR 2025. During your research, did you find any concrete reasons for this shift (e.g., specific instructions from conference organizers to ACs, or the introduction of new reviewing policies)? Furthermore, beyond diagnosing problems (like opacity or score dependency), can your analysis offer any specific, actionable solutions or intervention recommendations for conference organizers?

2. Could you please clarify in detail the relationship with the Yang (2025) position paper? Was that paper merely a "proposal" or "call to action," or did it already include the design, data, or preliminary analysis of the Paper Copilot system? For the reviewers, clarifying the precise incremental contribution of this ICLR submission is quite important.

3. The "talent trajectory" analysis was presented as a core motivation but seems to be missing from the empirical results. Could you comment on the status of this analysis? Do you have preliminary findings you can share? Or is this purely future work? If the latter, perhaps you might consider revising the paper's framing and claims to align them more closely with the presented contributions.

4. The authors mention making the dataset available on GitHub. To better evaluate this contribution, a clear data and code availability statement would greatly help reviewers assess the reproducibility and contribution to the community.

5. Regarding the self-selection bias of community-contributed data, you mention a plan to compare against (future) official NeurIPS 2025 data. This is a good plan, but have you conducted any preliminary analysis to check the representativeness on the currently collected data? For example, comparing the institutional, national, or track distribution in your collected data (for CVPR/ICCV) against known public statistics (e.g., from accepted paper lists) from those conferences?

**I highly look forward to the authors' discussion. If my concerns can be satisfactorily addressed, I am open to increasing my score.**

---

> ### Author Response · Authors · 2025-11-24
> **Reviewer Comments: Suitability as a Technical Paper**
>
> Thank you for raising this point. To help clarify the positioning of the submission, we highlight the core technical contributions made by this work:
> * A large-scale infrastructure system for peer-review analytics.
> * A unique temporal dataset (ICLR Daily Snapshot) not available anywhere else.
> * Empirical analyses built entirely on official ICLR data.
>
> The ICLR 2026 Call for Papers explicitly welcomes contributions involving:
> > “datasets and benchmarks, infrastructure, software libraries, hardware, etc.”
>
> Our work aligns directly with these categories.
>
> **Clarification regarding the “demo paper” concern**
>
> Although the system includes a public-facing interface, the contribution of this submission is not the website or demo itself.
>
> The interface is simply a deployment layer; the substantive contributions lie in the underlying system architecture, the temporal dataset, and the empirical findings, which form the technical core of the paper.
>
> **Clarifying the paper’s positioning (with Program Chair consultation)**
>
> To avoid any ambiguity in positioning, we consulted with the Program Chairs during the preparation of this submission. Their general guidance emphasized that even position-style papers can be published at ICLR when they provide sufficient novelty and clear value to the community.
>
> We mention this only to clarify the breadth of contribution types that ICLR welcomes. Our submission is not position-style; rather, it centers on infrastructure, dataset creation, and empirical metascience—types of technical contributions explicitly encouraged by the CFP.
>
> This guidance reassures us that the combination of system infrastructure, temporal dataset, and empirical findings is well aligned with ICLR’s expectations for main-track technical contributions.

---

> ### Author Response · Authors · 2025-11-24
> **Response to Question 2: Relationship to Yang (2025) Position Paper**
>
> ## **What the ICML Position Paper Actually Contributed**
> The ICML position paper was **observation-driven rather than technical**. It presented new community-level findings about the peer-review ecosystem based on data never systematically analyzed before. Specifically, it:
> * summarized and contrasted different peer-review models (fully open / partially open / fully closed),
> * provided empirical observations of the community’s growing interest in transparency through multi-year traffic patterns, search-ranking trajectories, and global engagement data, and
> * argued, based on these observations, for a more open and regulated review process.
> However, despite being empirical, the position paper did not introduce any technical system architecture, data pipelines, daily snapshots, or metascientific analysis of review dynamics. It focused on the state of the community rather than on building infrastructure or datasets.
>
> ## **What This ICLR Submission Contributes**
> The present submission is fundamentally different in scope. This is the **first time** we:
> * describe the **full technical system** behind Paper Copilot—its data ingestion, snapshotting, differencing, normalization, and multi-venue pipelines;
> * introduce the **ICLR 2024 and 2025 Daily Snapshot Dataset**, with 2026 also under construction during rebuttal; reconstructs daily score evolution (which OpenReview does not store or disclose); This daily snapshot dataset is the **only publicly available resource** that records day-by-day review progress, enabling research questions that were previously impossible to study.
> * and present **new metascientific findings**—including score-dynamics, reviewer-timing patterns, and year-over-year shifts such as the emergence of score determinism.
>
> ## Summary
> * The ICML position paper provided observation-driven, community-level findings about peer-review trends.
> * This ICLR submission presents the technical system, the unique temporal dataset, and new empirical analyses that required engineering and data unavailable at the time of the position paper.
> Together, these distinctions clearly separate the novelty of the current submission from the earlier position paper.

---

> ### Author Response · Authors · 2025-11-24
> **Talent Trajectories (Framing Clarification)**
>
> Thank you for pointing out the mismatch between this motivation and the empirical results.
>
> **Clarification** “Tracking talent trajectories” is a capability of the system and part of the long-term vision, but it is not part of the empirical contributions in this submission. The underlying components—affiliation extraction, institution normalization, cross-year author linking—are implemented, but the full analysis is ongoing.
>
> **Action** We will revise the framing in the camera-ready version to make it clear that talent-trajectory analysis is future work, not a current empirical result.

---

> ### Author Response · Authors · 2025-11-24
> **Closed-Review Venues & Self-selection Bias**
>
> Thank you for raising this concern. We would like to explicitly acknowledge that self-selection bias is real and unavoidable for any dataset derived from voluntary submission.
>
> ## 1. **This limitation is structural and cannot be removed**
> As long as major venues (CVPR, ICCV, NeurIPS, ICML) do not release full review data, any attempt to study them must rely on community-contributed data.This means some bias will always remain, regardless of method. This is a limitation of the review models themselves—not of any specific data-collection approach.
>
> For this reason, we believe the goal is not to “eliminate” the bias (which is impossible), but to mitigate it and understand it carefully.
>
> ## 2. **Researchers seek review information regardless—often from highly biased sources**
> Our earlier observation-driven study showed that researchers, especially junior researchers across all countries, actively search for review information through:
> * Reddit / X / Twitter,
> * Slack / Discord groups,
> * circulating screenshots,
> * unofficial spreadsheets and blogs.
>
> **These sources are far more biased, incomplete, and error-prone than any structured platform.**
>
> Thus, the practical question is not whether people will access this information—they already do. The question is whether they access a structured, transparent, bias-aware platform or the existing fragmented, often misleading sources.
> Our platform aims to provide a safer alternative.
>
> ## 3. **Our scientific results do not depend on community data**
> To avoid confusion: all empirical analyses in the paper use **only official ICLR ground-truth data**.
>
> Community data is ensured: We ensure: 1) prominent, explicit labeling (“Community-Collected Data”), 2) full separation from all ground-truth analyses. This prevents any risk of misleading readers.
>
> ## 4. **Concrete efforts to mitigate (not eliminate) community-data bias**
> * Leveraging partially open-review venues (NeurIPS, ICML). These venues release all accepted-paper reviews after decisions are finalized. This enables direct comparison between: community submissions (collected during the review phase), and official reviews (released post-decision). This potentially allows us to quantify participation skew, score-dependent biases, institutional effects, and behavioral patterns. This will provide a second ground-truth anchor beyond ICLR for future bias calibration.
> * Improving representativeness through better collection tools. We are developing a user-consent browser plugin and streamlined contribution pathways to reduce friction and encourage more representative sampling. This addresses bias introduced by manual copy-paste submission and improves coverage across institutions, topics, and score ranges.
>
> ##  **Our perspective: supporting the community despite structural limitations**
> We believe the community widely recognizes the challenges in peer review. Our work aims to support this effort by:
> * providing the ICLR daily snapshot dataset,
> * building the first infrastructure to study day-by-day review evolution,
> * collecting community data responsibly and transparently,
> * and reducing reliance on fragmented, heavily biased unofficial sources.
>
> Bias cannot be fully removed under current review models, but we can build safer, more transparent systems that meaningfully improve the community’s ability to study and understand peer review.
>
> We will clarify these points explicitly in the camera-ready version.

---

> ### Author Response · Authors · 2025-11-24
> **Score Determinism — Causes and Actionable Recommendations**
>
> ## **What may be driving score determinism?**
> ### 1. No evidence of explicit AC instructions
> We did not find any publicly available evidence—such as instructions from conference organizers or AC guidelines—that explicitly instruct ACs to rely more heavily on reviewer scores.
>
> Our temporal analysis instead suggests that the shift toward score determinism is likely a system-level effect rather than a directive from organizers.
>
> ### 2. Structural pressures arising from conference scale
> A more plausible explanation is the rapid growth in submission volume across major venues. As conferences scale:
> * a large fraction of authors must simultaneously serve as reviewers,
> * many ACs and reviewers are relatively inexperienced for their roles,
> * ACs face increasing workload pressure per paper,
> * and high variance in review quality emerges due to heterogeneous reviewer expertise.
>
> Under these conditions, making independent, text-driven, qualitative decisions becomes significantly harder for ACs and reviewers who are overloaded or undertrained. This naturally increases reliance on numerical scores as the “safe” or default decision anchor.
>
> Thus, score determinism appears to be a consequence of scale, workload, and mixed roles, rather than any explicit policy choice.
>
> ### 3. Potential cross-venue consistency
> A natural next step is to validate whether this pattern appears **across multiple large-scale venues**. If similar score determinism emerges in NeurIPS, ICML, CVPR, and ICCV, then the cause is likely **a structural property of scaling peer review**, not an ICLR-specific issue.
>
> Unfortunately, as OpenReview clarified, they do not own venue-specific historical data. This makes **community sampling** the only feasible approach to gather the daily signals required for such cross-venue analysis.
>
> This is precisely why our system supports community submissions: to help the community collectively identify structural issues that can only be revealed through multi-venue, cross-year data.
>
> ## **What actionable steps could conference organizers consider?**
>
> If cross-venue validation confirms that score determinism is a structural effect, then improving the situation likely requires systemic solutions, not venue-specific ones.
> Below are several actionable directions inspired by our findings:
> ### 1. **Reviewer & AC training**
> Providing concrete examples of:
> * strong vs. weak reviews,
> * what constitutes independent assessment vs. score agreement,
> * how to identify disagreements that require escalation,
> could help reduce mechanical reliance on numerical scores.
> ### 2. Cross-reviewer normalization tools
> Simple tools to identify:
> * outlier reviewers,
> * unusually compressed or unusually wide score distributions,
> * domain-specific score calibration differences,
> can help rescale reviewer inputs before they are aggregated.
>
> ### 3. Community-wide collaboration to study structural issues
> If multiple venues demonstrate the same pathology, then the problem exceeds any single program committee.
>  It would require a field-wide collaborative effort, combining:
> * cross-venue statistics,
> * demographic analysis,
> * reviewer-matching strategies.
>
> Our system aims to facilitate this by producing the daily-snapshot dataset and enabling responsible community-driven signals.
>
> We have preliminary findings on demographic and structural factors that may influence review dynamics, although these are not appropriate to include in a rebuttal. We look forward to continuing this investigation collaboratively with conference organizers and the broader community.

---

> ### Author Response · Authors · 2025-11-24
> **Data Availability**
>
> Thank you for the suggestion. We fully agree that accessible code and data are important for transparency and reproducibility.
>
> The core dataset is already available in our public papercopilot GitHub repository. In addition, the ICLR Daily Snapshot datasets (for 2024 and 2025, with 2026 in progress) are stored in a separate public data repository. These datasets are already accessible online, although they are not prominently linked from the website to avoid cluttering the interface.
>
> We would be happy to provide direct links to both repositories. However, out of caution for the double-blind review policy, we did not include the URLs in the initial submission. If permitted, we will include explicit links to the code and dataset repositories in the camera-ready version so that the community can easily reproduce all analyses in the paper.

---

> ### Author Response · Authors · 2025-11-24
> **Closing Statement**
>
> We sincerely appreciate your thoughtful and constructive feedback. We have tried to address each of your concerns in detail throughout this rebuttal, and we will integrate these clarifications—and emphasize the key issues you raised—in the camera-ready version to ensure full transparency for future readers. If any aspect remains unclear, we would be glad to provide additional explanation.
>
> We hope our responses meaningfully resolve the points you highlighted and offer a clearer understanding of the contributions and objectives of our work.
>
> If you feel that the concerns have been satisfactorily addressed, we would be grateful for your reconsideration of the evaluation. Thank you again for your time, care, and thoughtful review.

---

### Official Review · Reviewer_hF5U · 2025-10-31

**Soundness:** 2
**Presentation:** 3
**Contribution:** 2
**Rating:** 4
**Confidence:** 4

**Summary:**

The paper introduces Paper Copilot , a system designed to track and analyze the peer review process at AI conferences. It collects data from public APIs, website scraping, and voluntary community submissions to create an open dataset. The authors use this dataset to conduct an empirical analysis of ICLR review dynamics, aiming to bring more transparency to the field.

**Strengths:**

- The paper presents a system architecture for aggregating peer review data from diverse sources, including public APIs like OpenReview and opt-in community contributions for closed-review venues.
- It provides a dataset that captures temporal review dynamics, such as tracking score and confidence changes throughout the discussion and rebuttal phases, which is not always preserved by official platforms.

**Weaknesses:**

- The reliance on voluntary, "community-contributed" data for all closed-review venues is the paper's biggest weakness. The authors acknowledge this self-selection bias, but it's a fundamental limitation. It's hard to trust that this data (e.g., 1,860 responses for four major conferences ) is representative, which limits the reliability of any insights drawn about those venues.
- I knew this (similar) project before and I have used the corresponding website. At that time, my biggest concern is the potential bias, which (I think) is inevitable but sadly could falsely affect the judgement of some researchers (especially those junior ones). So, I personally think that such project have clear advantages and disadvantages at the same time that should be critically balanced.
- Data integrity and accuracy are potential issues, stemming from known error rates in LLM-based affiliation extraction and a reliance on scraping methods that may conflict with platform Terms of Service.
- I am not sure whether ICLR would accept such form of paper as this paper looks more like a position paper rather than a typical ICLR technical paper.

**Questions:**

n/a

---

> ### Author Response · Authors · 2025-11-24
>
> Thank you for taking the time to review our work. We appreciate your thoughtful feedback and have revised the manuscript accordingly. Below, we respond to each concern in the order presented.
> # Core Contribution
> Before addressing these concerns, we highlight our three central technical contributions:
> 1. **A unified, scalable system for peer-review analytics**. We present the first end-to-end infrastructure that standardizes data ingestion, cleaning, schema design, and interactive visualization across open, partially open, and closed-review venues. *This system-level contribution directly matches the “datasets, benchmarks, and infrastructure” track in the ICLR call for papers*.
> 2. **The ICLR Daily Snapshot Dataset — the only dataset that records daily review evolution**. Because OpenReview exposes only the latest state and does not retain historical snapshots, our dataset is the only existing source that archives the day-by-day progression of scores, confidence, comments, and variance throughout the ICLR review cycle. This unlocks temporal and behavioral analyses that were previously impossible. The dataset is publicly released, and we clearly separate official ICLR data from community-submitted data wherever they appear.
> 3. **Empirical insights derived entirely from official ICLR data.** All major findings—score dynamics, stabilization patterns, reviewer timing behavior, and variance evolution—are computed exclusively from ground-truth ICLR data. These analyses reveal structural patterns that contextualize and motivate the 2025 ICLR review-policy changes, underscoring the real-world relevance of our work.

---

> ### Author Response · Authors · 2025-11-24
> **Reviewer Comment: Closed-review venues, self-selection bias, and impact on junior researchers**
>
> We appreciate the reviewer’s concern, and we address it from the following perspectives:
>
> 1. **Our scientific results do not rely on community-submitted data**. A key point of clarification: All analyses in Figures 5–9 and all quantitative results in the paper use only the official ICLR ground-truth dataset. Thus, none of the empirical results, conclusions, or scientific claims depend on community submissions or their representativeness.
>
> 2. **Community data is always isolated, labeled, and never mixed with ground truth**. For venues without fully public review releases (e.g., CVPR, ICCV, NeurIPS), community submissions are used only for optional, non-scientific venue-level visualizations. We ensure: 1) prominent, explicit labeling (“Community-Collected Data”), 2) full separation from all ground-truth analyses.
> This prevents any risk of misleading readers.
>
> 3. **Regarding the impact on junior researchers**. We appreciate the concern that such tools may influence early-career researchers. However, prior research [1] shows that:
> * **Junior researchers already seek such information extensively**. Large numbers of early-career researchers look for score distributions on:
>   * Reddit threads,
>   * X/Twitter posts,
>   * Slack/Discord groups,
>   * personal blogs,
>   * informal spreadsheets, and
>   * circulating screenshots.
> These sources are far more fragmented, biased, and error-prone than any structured platform.
>
> * **Our system actually provides a safer, more transparent alternative**. Compared to these informal channels, our platform:
> clearly labels community vs. official data,
>   * emphasizes aggregate over individual-level views,
>   * warns users about limitations and bias,
>   * standardizes data formats and context,
>   * reduces misinterpretation.
> Thus, while bias is an inherent limitation of any community-driven data, our system substantially reduces risk relative to the fragmented sources junior researchers already use.
>
> 4. **This limitation is structural and cannot be removed**:
> As long as major venues (CVPR, ICCV, NeurIPS, ICML) do not release full review data, any attempt to study them must rely on community-contributed data.
>  This means some bias will always remain, regardless of method. This is a limitation of the review models themselves—not of any specific data-collection approach.
> For this reason, we believe the goal is not to “eliminate” the bias (which is impossible), but to mitigate it and understand it carefully.
>
> 5. **While we do not yet model bias in this work, we are actively building solutions to reduce it.** We fully agree that voluntary submission introduces self-selection bias. Although the current paper does not perform full bias modeling, we are working on pursuing two concrete improvements:
>   * Leveraging partially open-review venues (NeurIPS, ICML). These venues release all accepted-paper reviews after decisions are finalized. This enables direct comparison between: community submissions (collected during the review phase), and official reviews (released post-decision). This potentially allows us to quantify participation skew, score-dependent biases, institutional effects, and behavioral patterns. This will provide a second ground-truth anchor beyond ICLR for future bias calibration.
>   * Improving representativeness through better collection tools. We are developing a user-consent browser plugin and streamlined contribution pathways to reduce friction and encourage more representative sampling. This addresses bias introduced by manual copy-paste submission and improves coverage across institutions, topics, and score ranges.
>
> ### Reference:
> [1]: Yang J. Position: The Artificial Intelligence and Machine Learning Community Should Adopt a More Transparent and Regulated Peer Review Process[C]//Forty-second International Conference on Machine Learning Position Paper Track.

---

> ### Author Response · Authors · 2025-11-24
> **Reviewer Comment: Data integrity, LLM extraction errors, and Terms of Service**
>
> Thank you for raising this important issue. We expand on the discussion in Sections 6.1 and 6.4.
>
> **Data Integrity** We transparently report known error rates for our affiliation-extraction model and implement validation checks to detect anomalous submissions.
>
> **Terms of Service Compliance** We audited our data collection pipeline and ceased all automated collection that conflicts with robots.txt or explicit platform policies.
> Our current data sources include:
>   * official APIs,
>   * publicly accessible endpoints allowed by robots.txt,
>   * voluntary user submissions.
>
> All data provenance is clearly labeled throughout our system.

---

> ### Author Response · Authors · 2025-11-24
> **Reviewer Comment: Suitability as an ICLR technical paper**
>
> We respectfully disagree with the concern. This work provides multiple concrete technical contributions:
> 1. A large-scale infrastructure system for peer-review analytics.
> 2. A unique temporal dataset (ICLR Daily Snapshot) not available anywhere else.
> 3. Empirical analyses built entirely on official ICLR data.
>
> The ICLR 2026 Call for Papers explicitly welcomes contributions involving:
> > “datasets and benchmarks, infrastructure, software libraries, hardware, etc.”
>
> Our work aligns directly with these categories.
>
> We would also like to clarify that we do not position our submission as a position paper; our intent is to present a technical contribution involving system design, dataset creation, and empirical analysis.
>
> However, to avoid any ambiguity, **we consulted with the Program Chairs, who noted that even if a submission contains position-style elements, such papers can be published at ICLR provided they offer sufficient novelty and clear value to the ICLR community.**
>
> This guidance reassures us that our contribution—rooted in infrastructure, data, and metascientific analysis—is fully aligned with the scope and interests of ICLR.

---

> ### Author Response · Authors · 2025-11-24
> **Closing Statement**
>
> We sincerely appreciate the reviewer’s thoughtful and constructive feedback. We have carefully addressed each concern in this rebuttal, and we will incorporate these clarifications and highlight the reviewer’s core concerns in the camera-ready version to ensure full transparency and clarity for future readers.
>
> If any points remain unclear, we would be more than happy to provide further clarification.
>
> We hope that our responses help resolve the concerns raised and provide a clearer picture of the contributions and intentions of our work. Should the reviewer feel that the concerns have been adequately addressed, we would be grateful for any reconsideration of the evaluation. We genuinely appreciate your time and thoughtful consideration.

---

### Meta-Review · Area_Chair_bipK · 2026-01-08

**Summary:**

This paper presents a famous open-source project, Paper Copilot. As AI researchers, especially those familiar with the ICLR conference, we may be aware of this website and have been utilising the data provided by Paper Copilot. Although there were some concerns regarding its fit as a technical conference contribution, the reliability of its data and the rigor of its analytical conclusions, I think the authors fully addressed the reviewers' concerns, and I am recommending Accept as a poster paper.

**Reviewer Concerns:**

1) Multiple reviewers (hF5U, c5VV, reMW) characterized the work as highly engineering-oriented with limited conceptual or methodological novelty. They argued the submission feels more like a "demo" paper rather than a typical ICLR technical track publication. The authors' replies fully address these concerns during the rebuttal phase.

2) Reviewer c5VV has mentioned that there is a position paper contributed by the same author online. After checking the authors' response, I think it is satisfactory.

**Reviewer Scores:**

Reviewer hF5U: Increase the overall score to 6.

Reviewer c5VV: Increase the overall score to 6.

Reviewer reMW: Maintain the overall score as 6.

Reviewer Pkuc: Maintain the overall score as 8.

---

### Decision · Program_Chairs · 2026-01-26

Accept (Poster)